# Cystathionine-β-synthase is essential for AKT-induced senescence and suppresses the development of gastric cancers with PI3K/AKT activation

Haoran Zhu[1,2†], Keefe T Chan[1,2†], Xinran Huang[1,2], Carmelo Cerra[1], Shaun Blake[1], Anna S Trigos[1,2], Dovile Anderson[3], Darren J Creek[3], David P De Souza[4], Xi Wang[5], Caiyun Fu[6], Metta Jana[1,2], Elaine Sanij[1,2,7,8,9,10], Richard B Pearson[1,2,9,11]*, Jian Kang[1,2]*

[1]Division of Cancer Research, Peter MacCallum Cancer Centre, Melbourne, Australia; [2]Sir Peter MacCallum Department of Oncology, University of Melbourne, Melbourne, Australia; [3]Monash Institute of Pharmaceutical Sciences, Victoria, Australia; [4]Metabolomics Australia, Bio21 Molecular Science and Biotechnology Institute, Victoria, Australia; [5]Department of Oncology, The People's Liberation Army No. 903rd Hospital, Hangzhou, China; [6]Zhejiang Provincial Key Laboratory of Silkworm Bioreactor and Biomedicine, College of Life Sciences and Medicine, Zhejiang Sci-Tech University, Hangzhou, China; [7]St Vincent's Institute of Medical Research, Melbourne, Australia; [8]Department of Clinical Pathology, University of Melbourne, Melbourne, Australia; [9]Department of Biochemistry and Molecular Biology, Monash University, Melbourne, Australia; [10]Department of Medicine, St Vincent's Hospital, University of Melbourne, Melbourne, Australia; [11]Department of Biochemistry and Molecular Biology, University of Melbourne, Melbourne, Australia

*For correspondence:
rick.pearson@petermac.org (RBP);
jian.kang@petermac.org (JK)

†These authors contributed equally to this work

Competing interest: The authors declare that no competing interests exist.

**Abstract** Hyperactivation of oncogenic pathways downstream of RAS and PI3K/AKT in normal cells induces a senescence-like phenotype that acts as a tumor-suppressive mechanism that must be overcome during transformation. We previously demonstrated that AKT-induced senescence (AIS) is associated with profound transcriptional and metabolic changes. Here, we demonstrate that human fibroblasts undergoing AIS display upregulated cystathionine-β-synthase (CBS) expression and enhanced uptake of exogenous cysteine, which lead to increased hydrogen sulfide ($H_2S$) and glutathione (GSH) production, consequently protecting senescent cells from oxidative stress-induced cell death. CBS depletion allows AIS cells to escape senescence and re-enter the cell cycle, indicating the importance of CBS activity in maintaining AIS. Mechanistically, we show this restoration of proliferation is mediated through suppressing mitochondrial respiration and reactive oxygen species (ROS) production by reducing mitochondrial localized CBS while retaining antioxidant capacity of transsulfuration pathway. These findings implicate a potential tumor-suppressive role for CBS in cells with aberrant PI3K/AKT pathway activation. Consistent with this concept, in human gastric cancer cells with activated PI3K/AKT signaling, we demonstrate that CBS expression is suppressed due to promoter hypermethylation. CBS loss cooperates with activated PI3K/AKT signaling in promoting anchorage-independent growth of gastric epithelial cells, while CBS restoration suppresses the growth of gastric tumors in vivo. Taken together, we find that CBS is a novel regulator of AIS and a potential tumor suppressor in PI3K/AKT-driven gastric cancers, providing a new exploitable metabolic vulnerability in these cancers.

### Editor's evaluation

This paper describes a new mechanism of metabolic escape from senescence. In cells undergoing senescence induced by AKT, the enzyme cystathionine β-synthase (CBS) maintains viability in the senescent state. Suppressing CBS results in senescence escape and continued proliferation, through a mechanism involving changes in mitochondrial metabolism.

## Introduction

Hyperactivation of oncogenic pathways such as RAS/ERK or PI3K/AKT can cause cellular senescence in non-transformed cells, termed oncogene-induced senescence (*Serrano et al., 1997*; *Zhu et al., 2020*). In addition to the well-studied RAS-induced senescence (RIS), we and others have demonstrated that hyperactivation of PI3K/AKT signaling pathway causes a senescence-like phenotype, referred to as AKT-induced senescence (AIS) or PTEN loss-induced cellular senescence (*Alimonti et al., 2010*; *Astle et al., 2012*; *Chan et al., 2020*; *Jung et al., 2019*). AIS is characterized by the common senescence hallmarks including cell cycle arrest, a senescence-associated secretory phenotype (SASP), global transcriptional changes, and metabolic hyperactivity (*Chan et al., 2020*). Distinct from RIS, AIS does not display either p16 upregulation, a DNA damage response or senescence-associated heterochromatin foci. Instead, AIS is associated with elevated p53 expression through increased mTORC1-dependent translation and reduced human double minute 2 (HDM2) dependent destabilization (*Astle et al., 2012*). Disruption of the critical mechanisms that regulate maintenance of oncogene-induced senescence can lead to tumorigenesis (*Braig et al., 2005*; *Chen et al., 2005*; *Collado et al., 2005*). Therefore, understanding the molecular mechanisms that regulate AIS and how they are subverted will provide opportunities to identify therapeutic strategies for suppressing PI3K/AKT-driven cancer development.

We identified 98 key regulators in a whole-genome siRNA AIS escape screen and validated a subset of these genes in the functional studies to confirm their role in AIS maintenance (*Chan et al., 2020*). Intriguingly, 11 genes were associated with the regulation of metabolism, suggesting that an altered metabolism could be integral for maintaining AIS. In particular, the cystathionine-β-synthase (*CBS*) was ranked as one of the top metabolic gene candidates with loss of expression leading to AIS escape (*Chan et al., 2020*), but how it does so is not known.

CBS is an enzyme involved in the transsulfuration metabolic pathway. CBS converts homocysteine (Hcy), a key metabolite in the transmethylation pathway, to cystathionine which is subsequently hydrolyzed by cystathionine gamma-lyase (CTH) to form cysteine, the crucial precursor for GSH production (*Figure 1*). CBS also catalyzes the production of $H_2S$, a diffusible gaseous transmitter that modulates mitochondrial function and cellular bioenergetics (*Szabo et al., 2013*; *Szabo et al., 2014*; *Módis et al., 2014*), exerts antioxidant effects through inhibition of reactive oxygen species (ROS) generation and lipid peroxidation (*Wen et al., 2013*), and stimulates antioxidant production via sulfhydration of key proteins involved in antioxidant defense such as Keap1 and p66Shc (*Paul and Snyder, 2012*; *Yang et al., 2013*). Thus, CBS acts through control of Hcy, $H_2S$, and GSH metabolism and exerts diverse biological functions including regulating DNA methylation, mitochondrial respiration, and redox homeostasis (*Zhu et al., 2018*).

Aberrant CBS expression and/or activity contributes to a wide range of diseases including hyperhomocysteinemia (*Kruger, 2017*) and cancer (*Zhu et al., 2018*). CBS plays a complex role in cancer pathogenesis having purported tumor-promoting and -suppressive roles. Activation of CBS promoted tumor growth in colon (*Phillips et al., 2017*; *Szabo et al., 2013*), ovarian (*Bhattacharyya et al., 2013*), breast (*Sen et al., 2015*), prostate (*Liu et al., 2016*), and lung cancers (*Szczesny et al., 2016*), whereas loss of CBS in glioma cells increased tumor volume in vivo (*Takano et al., 2014*). In addition, the function of CBS in liver cancer remains inconclusive with conflicting reports of both tumor-promoting (*Jia et al., 2017*; *Yin et al., 2012*) and -suppressive roles (*Kim et al., 2009*). These studies underscore the context-dependent roles of CBS in cancer development.

In this study we explored the molecular mechanisms underpinning CBS's role in maintaining AIS and how the loss of CBS promotes AIS escape. The requirement of CBS for the maintenance of AIS implicates it as a putative tumor suppressor during PI3K/AKT pathway-driven tumorigenesis. To gain insight into this, we further characterized the expression level of CBS in gastric cancer tissue samples

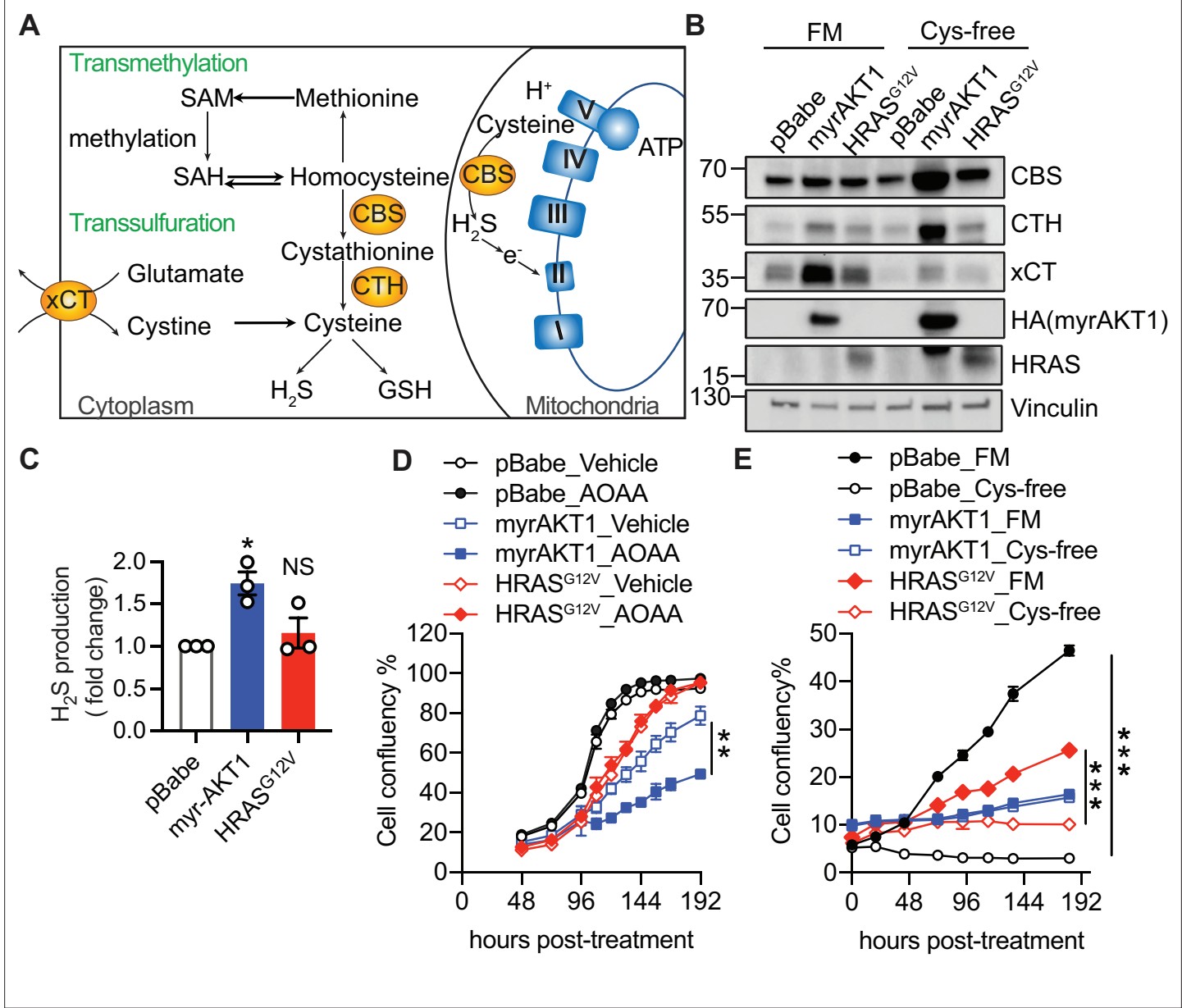

**Figure 1.** Cystathionine-β-synthase (CBS) expression and transsulfuration pathway activity are elevated in AKT-induced senescence. (**A**) Schematic diagram illustrating that the cytoplasmic localized CBS regulates transmethylation and transsulfuration metabolic pathways, and mitochondrial localized CBS regulates oxidative phosphorylation. (**B**) BJ3 human skin fibroblasts expressing telomerase reverse transcriptase (BJ-TERT) cells were transduced with pBabe, myrAKT1, or HRAS$^{G12V}$. On day 6 post-transduction the cells were plated in either full culture medium containing 100 µM cysteine (FM) or cysteine-deficient medium (Cys-free). Western blot analysis was performed on day 10 post-transduction. Vinculin was probed as a loading control. Representative of n=3 experiments. (**C**) Hydrogen sulfide (H$_2$S) production was measured by AzMC on day 14 post-transduction. Fold changes over pBabe control are presented as mean ± SEM (n=3). One sample t-test compared to the hypothetical value 1.0 was performed (NS, not significant; *p<0.05). (**D**) Cells were treated with aminoxyacetate (AOAA) 30 µM on day 5 post-transduction. Cell confluency measured by IncuCyte is presented as mean ± SEM (n=3). (**E**) Cells were cultured in the conditions as described in (**B**). Cell confluency measured by IncuCyte is presented as mean ± SEM (n=3–5). Statistical significance at the last time point in (**D**) and (**E**) was determined by unpaired t-test (**p<0.01; ***p<0.001).

The online version of this article includes the following source data and figure supplement(s) for figure 1:

**Source data 1.** Unedited immunoblots of *Figure 1B*.

**Figure supplement 1.** Cystathionine-β-synthase (CBS) expression and transsulfuration pathway activity are elevated in AKT-induced senescence.

**Figure supplement 1—source data 1.** Unedited immunoblots of *Figure 1—figure supplement 1B*.

and cells and sought to define the functional significance of CBS loss in the context of activated PI3K/AKT signaling-driven gastric cancer development.

## Results

### CBS expression and transsulfuration pathway activity are elevated in AIS

To investigate the mechanisms by which CBS contributes to AIS maintenance, we first evaluated CBS expression and activity in several non-transformed cells with hyperactivated AKT. An increase of CBS protein expression was observed in BJ3 human skin fibroblasts expressing telomerase reverse transcriptase (BJ-TERT) (*Figure 1B*) and IMR90 human fetal lung fibroblasts (*Figure 1—figure supplement 1A*) overexpressing myristoylated (myr)-AKT1. In BJ-TERT and human mammary epithelial cells (HMEC), overexpressing AKT1$^{E17K}$, a clinically relevant activated mutant form of AKT1 in multiple cancer types including breast cancer and ovarian cancer (*Carpten et al., 2007*), also enhanced CBS protein expression (*Figure 1—figure supplement 1B*). However, AKT hyperactivation did not affect *CBS* mRNA expression (*Figure 1—figure supplement 1C*), suggesting a post-transcriptional regulatory mechanism underpinning increased CBS protein expression.

We hypothesized that the increased CBS expression in AKT-hyperactivated cells was associated with upregulation of the transsulfuration pathway activity and cysteine metabolism. We thus examined the expression levels of CTH, a key enzyme in the transsulfuration pathway and xCT, the Xc- amino acid antiporter responsible for the uptake of cystine (an oxidized form of cysteine) (*Figure 1A*). Both CTH and xCT were upregulated in AIS cells compared to proliferating control cells, suggesting an elevated cysteine synthesis via the transsulfuration pathway and cysteine uptake (*Figure 1B*). In contrast, senescent cells expressing HRAS$^{V12}$ cells exhibited a moderate increase of CBS and CTH expression levels. The expression level of xCT was also slightly upregulated during RIS, albeit to a lesser extent than during AIS, in line with the finding that upregulation of xCT facilitates RAS-mediated transformation (*Lim et al., 2019*).

To assess transsulfuration pathway activity, we measured $H_2S$ production. A significant increase in transsulfuration pathway activity was observed in BJ-TERT fibroblasts upon AKT but not HRAS hyperactivation (*Figure 1C*), suggesting that activation of transsulfuration pathway is a specific cellular response to constitutive activation of AKT. Inhibition of $H_2S$ production by aminoxyacetate (AOAA) (*Szabo, 2016*) impaired cell proliferation (*Figure 1D*) and increased SA-βGal activity (*Figure 1—figure supplement 1D*) of BJ-TERT cells overexpressing myrAKT1. This result suggests that $H_2S$, the major metabolite downstream of the transsulfuration pathway, has a protective effect on AIS cells although the actions of AOAA on other PLP-dependent enzymes cannot be excluded (*Asimakopoulou et al., 2013*; *Hellmich et al., 2015*; *Szabo et al., 2013*). Cysteine starvation has been reported to induce necrosis and ferroptosis in cancer cells (*Chen et al., 2017*). Since the transsulfuration pathway mediates de novo cysteine synthesis, an increase in transsulfuration pathway activity may support the survival of AIS cells upon cysteine limitation. Consistent with our hypothesis, cysteine deprivation potently increased the expression of CBS and CTH in AIS cells (*Figure 1B*) and did not affect the survival of AIS cells (*Figure 1E*), indicating that increased cysteine level in AIS cells due to elevated transsulfuration pathway activity is critical for cell viability.

### Depletion of CBS promotes escape from AIS

While our results suggest a protective role of transsulfuration pathway for the survival of AIS cells, our AIS siRNA screen showed *CBS* loss enabled cells to escape from AIS evidenced by an increase in cell numbers. To validate the function of CBS in AIS maintenance, we depleted *CBS* using two independent small hairpin RNAs (shRNAs) in AIS cells (*Figure 2A*). Loss of CBS in AIS cells significantly decreased the proportion of cells with SA-βGal activity, increased EdU incorporation, and enhanced colony formation, demonstrating an essential role of CBS in AIS maintenance (*Figure 2B*). To further confirm the on-target specificity of the knockdown, we generated BJ-TERT cells expressing a doxycycline-inducible *CBS* shRNA and an shRNA-resistant 4-OHT-inducible estrogen receptor (ER)-tagged CBS fusion (*Figure 2C* and *Figure 2—figure supplement 1A*). Upon expressing myrAKT1, these cells underwent AIS, as indicated by a significant increase in SA-βGal-positive cells and decrease in EdU-positive cells and, consistent with the finding in the AIS escape siRNA screen, CBS depletion induced

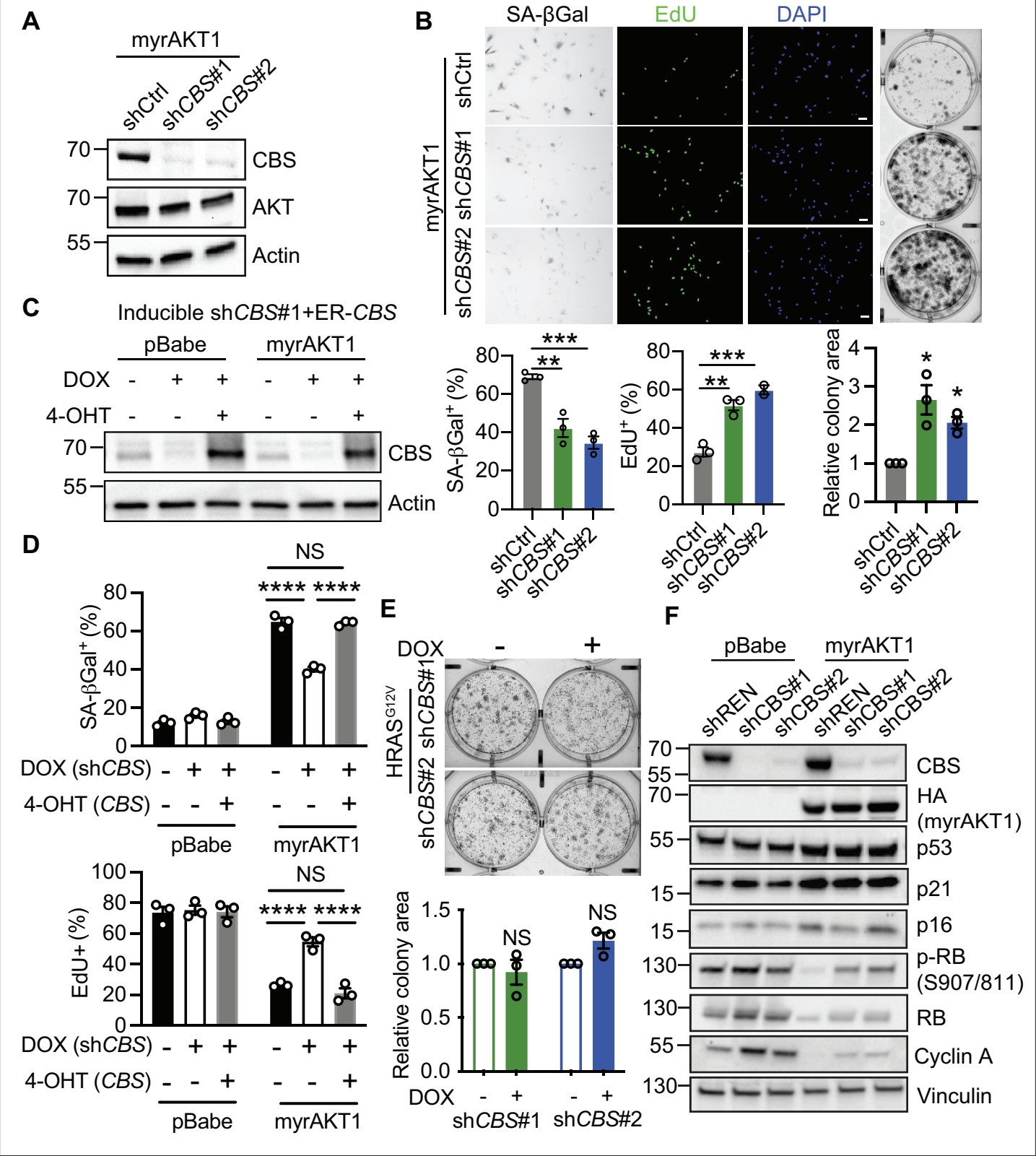

**Figure 2.** Depletion of cystathionine-β-synthase (CBS) promotes escape from AKT-induced senescence. (**A and B**) BJ3 human skin fibroblasts expressing telomerase reverse transcriptase (BJ-TERT) cells expressing myrAKT1 were transduced with pGIPZ-sh*CBS* or non-silencing small hairpin RNA (shRNA) control (shCtrl) on day 6 post-transduction of myrAKT1. (**A**) Western blot analysis was performed on day 8 post-transduction of shRNA. Representative of n=2 experiments. (**B**) Images and quantification of the percentage of cells with positive staining for SA-βGal activity and EdU incorporation on day 8

*Figure 2 continued on next page*

*Figure 2 continued*

post-transduction of shRNA, as well as colony formation assay on day 14 post-transduction of shRNA. Data is presented as mean ± SEM (n=3). One-way analysis of variance (ANOVA) with Holm-Šídák's multiple comparisons was performed (**p<0.01; ***p<0.001). Relative colony area normalized to shCtrl group is presented as mean ± SEM (n=3) and one sample t-test compared to the hypothetical value 1.0 was performed (**C–D**) BJ-TERT cells expressing doxycycline-inducible *CBS* shRNA#1 and 4-OHT-inducible CBS were transduced with pBabe or myrAKT1, treated with ±doxycycline (1 µg/ml)±4 OHT (20 nM) on day 5 post-transduction and analyzed on day 14 post-transduction. (**C**) Western blot analysis of CBS expression. Actin was probed as a loading control. (**D**) The percentage of cells with positive staining for SA-βGal activity or EdU proliferation marker incorporation is presented as mean ± SEM (n=3). Two-way ANOVA with Holm-Šídák's multiple comparisons was performed (NS, not significant; ****p<0.0001). (**E**) BJ-TERT cells expressing doxycycline-inducible *CBS* shRNA were transduced with HRAS$^{G12V}$, treated with doxycycline (1 µg/ml) on day 5 post-transduction and colony formation assay analyzed on day 14 post-transduction. Relative colony area normalized to doxycycline-untreated group is presented as mean ± SEM (n=3) and one-sample t-test compared to the hypothetical value 1.0 was performed (NS, not significant). (**F**) BJ-TERT cells expressing doxycycline-inducible sh*CBS* or control shREN were transduced with pBabe or myrAKT1 and then treated with doxycycline (1 µg/ml) on day 5 post-transduction. Western blot analysis was performed on day 14 post-transduction. Vinculin was probed as a loading control. Representative of n=2–4 experiments.

The online version of this article includes the following source data and figure supplement(s) for figure 2:

**Source data 1.** Unedited immunoblots of *Figure 2A*.

**Source data 2.** Unedited immunoblots of *Figure 2C*.

**Source data 3.** Unedited immunoblots of *Figure 2F*.

**Figure supplement 1.** Depletion of cystathionine-β-synthase (CBS) promotes escape from AKT-induced senescence.

**Figure supplement 1—source data 1.** Unedited immunoblots of *Figure 2—figure supplement 1A* (A) and 1D (B).

**Figure supplement 1—source data 2.** Unedited immunoblots of *Figure 2—figure supplement 1E*.

by doxycycline decreased SA-βGal activity, and increased EdU incorporation in AIS cells (*Figure 2D*). Importantly, simultaneously expressing ER-CBS prevented senescence escape of CBS-depleted cells, confirming the on-target specificity of the knockdown and the critical role of CBS in maintaining AIS (*Figure 2D* and *Figure 2—figure supplement 1B*). Modulation of CBS expression in proliferating control cells did not affect the percentage of SA-βGal- and EdU-positive cells (*Figure 2D*), suggesting that the effect of CBS depletion on cell proliferation is specific for AIS cells. Similar to the findings in BJ-TERT cells, AKT1 hyperactivation also caused senescence in IMR-90 lung fibroblasts (*Figure 2—figure supplement 1C*). *CBS* knockdown in AIS cells significantly suppressed SA-βGal staining and enhanced EdU incorporation but not in proliferating cells. Knockdown of *CBS* in BJ-TERT cells with constitutive RAS activation did not affect colony formation, suggesting CBS has a specific regulatory role for AIS but not RIS maintenance (*Figure 2E* and *Figure 2—figure supplement 1D*).

To determine the mechanisms by which CBS depletion causes escape, we examined the impact on key senescence hallmarks. While loss of CBS released AIS cells from cell cycle arrest, knockdown of *CBS* did not significantly change the mRNA and protein expression level of several key SASP-related genes including *IL1A, IL1B, IL6,* and *IL8,* which are upregulated during AIS (*Astle et al., 2012*; *Chan et al., 2020*; *Figure 2—figure supplement 1E and F*). Given the p53 and Retinoblastoma protein (Rb) pathways predominantly control senescence-mediated cell cycle arrest, we examined signaling downstream of AKT activation in the presence and absence of CBS (*Figure 2F*). Consistent with our previous findings, p53 and its downstream target p21 were upregulated during AIS, but depletion of CBS had no effect on these levels. Furthermore, total Rb, a key regulator of the G1/S phase transition, and its inhibitory phosphorylated form at serine 807/811 were markedly suppressed in AIS and partially rescued upon CBS knockdown. Cyclin A, which mediates S to G2/M phase cell cycle progression, was also upregulated upon depleting CBS in AIS cells. These results demonstrate that CBS depletion can restore the proliferation of cells that have undergone AIS in a p53-independent manner.

## Depletion of CBS in AIS cells does not affect cysteine and GSH abundance in cysteine-replete conditions

CBS is the key enzyme regulating the transsulfuration and transmethylation pathways. By analysis of the data from the AIS escape siRNA screen, we found that except *CBS*, siRNA knockdown of other genes involved in the transsulfuration and transmethylation pathway did not significantly affect AIS cell numbers (robust Z score <2, *Figure 3—figure supplement 1A*). Therefore, it is likely that AIS escape in cysteine-replete conditions upon CBS loss is through a transsulfuration/transmethylation pathway-independent mechanism. In addition, knockdown of *CBS* did not affect the expression levels

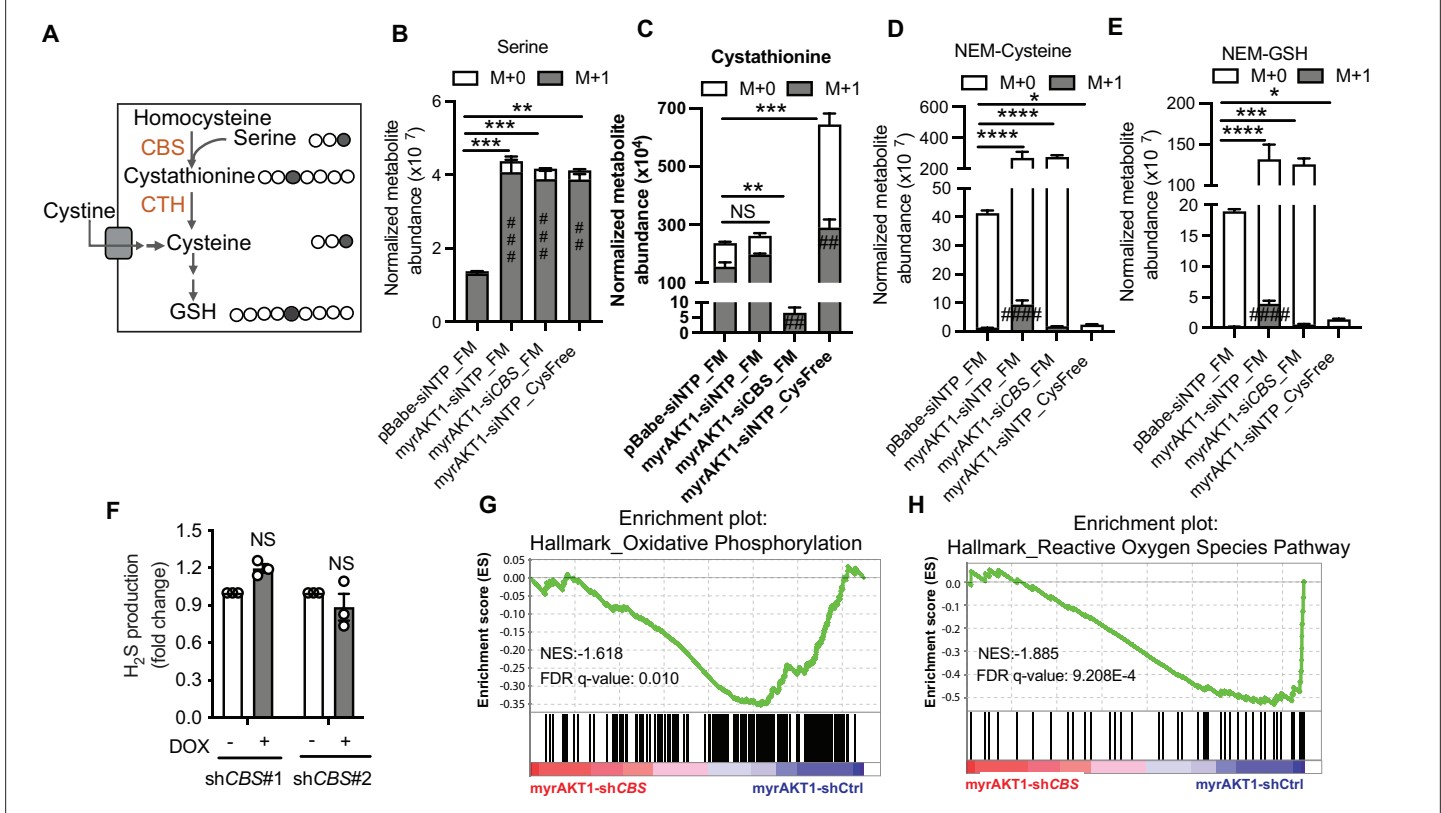

**Figure 3.** Depletion of cystathionine-β-synthase (CBS) in AKT-induced senescence (AIS) cells does not affect cysteine and glutathione (GSH) abundance in cysteine-replete conditions. (**A**) Schematic of [3–13C] serine isotope tracing. Gray circles indicate 13C carbon atoms. Clear circles indicate unlabeled carbon atoms. (**B–E**) BJ3 human skin fibroblasts expressing telomerase reverse transcriptase (BJ-TERT) cells were transduced with pBabe or myrAKT1. After 6 days cells were transfected with either *CBS* siRNA (siCBS) or control siRNA (siOTP). On day 3 post-siRNA transfection, cells were cultured in either full medium (FM) or cysteine-deficient medium (Cys-Free) for 48 hr. Six hours before harvest, the culture medium was replaced with the basal isotope labeling medium containing 400 μM [3–13C] serine. The thiol redox metabolome was assessed by targeted liquid chromatography mass spectrometry (LC/MS). The abundances of labeled and unlabeled metabolites including (**B**) serine, (**C**) cystathionine, (**D**) NEM-Cys, and (**E**) NEM-GSH normalized with cell number are presented as mean ± SEM (n=4). Statistical significance was determined by one-way analysis of variance (ANOVA) with Holm-Šídák's multiple comparisons (for the total metabolite levels, *p<0.05; **p<0.01; ***p<0.001; ****p<0.0001; and for the M+1 metabolite levels, ##p<0.01; ###p<0.001; ####p<0.0001 compared to pBabe-siNTP_FM cells). (**F**) BJ-TERT cells expressing doxycycline-inducible shCBS were transduced with myrAKT1, and treated with doxycycline (1 μg/ml) on day 5 post-transduction. Hydrogen sulfide (H2S) production was measured by AzMC on day 14 post-transduction. Fold changes over doxycycline-untreated group are presented as mean ± SEM (n=3). One sample t-test compared to the hypothetical value 1.0 was performed (NS, not significant). (**G and H**) Gene set enrichment analysis of RNA-seq data showing downregulation of hallmark of oxidative phosphorylation and reactive oxygen species pathways in myrAKT1-shCBS cells compared with myrAKT1-shCtrl cells.

The online version of this article includes the following figure supplement(s) for figure 3:

**Figure supplement 1.** Depletion of cystathionine-β-synthase (CBS) in AKT-induced senescence (AIS) cells does not affect cysteine and glutathione (GSH) abundance in cysteine-replete conditions.

of CTH and xCT when BJ-TERT cells were grown in cysteine-replete culture medium (*Figure 3—figure supplement 1B*).

To explore metabolic alterations that are associated with CBS-mediated AIS maintenance, we performed gas chromatography mass spectrometry (GC/MS)-based untargeted metabolomics. AIS (myrAKT1-shCtrl), AIS-escaped (myrAKT1-sh*CBS*), and control proliferating cells (pBabe-shCtrl) displayed distinct metabolic profiles, as indicated by principal component analysis (*Figure 3—figure supplement 1C and D*). AIS cells showed increased abundance of serine, glycine, and glutamate, the metabolites involved in transsulfuration pathway and GSH synthesis (*Figure 3—figure supplement 1F*). Loss of CBS did not affect abundance of these metabolites but resulted in an increased level of Hcy, suggesting loss of CBS causes an accumulation of the upstream substrate Hcy (*Figure 3— figure supplement 1F*). To further determine the activity of the transsulfuration pathway in AIS cells and the impact of CBS loss, we performed a stable isotope tracing assay using serine labeled at the

third carbon position ([3–13C] L-serine) followed by liquid chromatography mass spectrometry (LC/MS) after thiol derivatization with *N*-ethylmaleimide (*Figure 3A*). This tracer has been reported to incorporate into the cellular GSH pool via transsulfuration-derived cysteine (*Zhu et al., 2019*). We replaced all the serine in the culture medium with [3–13C] L-serine. After 6 hr of labeling, a substantial fraction of [3–13C] L-serine was detected intracellularly and in the cystathionine pool in proliferating (pBabe-siOTP), AIS (myrAKT1-siOTP), and AIS-escaped (myrAKT1-si*CBS*) cells (*Figure 3B and C*). We did not detect [3–13C] L-serine incorporation into cysteine and GSH in proliferating cells, possibly due to the short time period of metabolic labeling (*Figure 3D and E*). However, AIS cells displayed a small but significant fraction of 13C labeled cysteine and GSH along with a significant increase of total levels of serine, cysteine, and GSH (*Figure 3B-E*), indicating upregulation of transsulfuration pathway activity during AIS. Consistent with the role of CBS in catalyzing de novo cystathionine synthesis, a significant decrease of cystathionine abundance was observed in CBS-depleted AIS cells. Notably, the abundance of cysteine and GSH was not affected by CBS depletion (*Figure 3D and E*) and neither was $H_2S$ production (*Figure 3F*). We hypothesized that CBS-depleted cells maintained the cysteine and GSH pools via increase of cysteine uptake from the culture medium. Indeed, deprivation of cysteine from the medium markedly diminished the intracellular cysteine and GSH abundance in AIS cells (*Figure 3D and E*). On the other hand, increase of cystathionine was observed in the cysteine-depleted conditions (*Figure 3C*), possibly attributed to a marked upregulation of CBS expression observed in AIS cells after cysteine deprivation (*Figure 1B*). This result suggests that cells enhance CBS-mediated transsulfuration pathway activity in response to cysteine deficiency. If cells rely on exogenous cysteine to maintain cysteine and GSH pools, blocking cysteine import would impair GSH synthesis and antioxidant capacity and consequently reduce cell viability. Consistent with this hypothesis, the viability of both proliferating cells and AIS cells was decreased after treatment with erastin, an inhibitor of cystine-glutamate antiporter Xc-, and further suppressed by *CBS* knockdown (*Figure 3—figure supplement 1F*). Taken together, these results conclusively demonstrate that exogenous cysteine is the major source for GSH synthesis and AKT overexpression increases cysteine import and the subsequent GSH abundance. As the loss of CBS does not affect intracellular cysteine and GSH levels, senescence escape upon CBS depletion is mediated by the mechanisms independent of the transsulfuration pathway.

To investigate the transsulfuration pathway-independent molecular mechanisms underlying CBS-mediated AIS maintenance, we characterized the transcriptomic changes upon depleting CBS during AIS. Differential gene expression analysis of AIS-escaped cells (AIS-sh*CBS*) compared with AIS cells (AIS-shCtrl) revealed 404 genes were significantly upregulated (adjusted p-value < 0.05, Log$_2$FC > 1) and 181 genes significantly downregulated (adjusted p-value < 0.05, Log$_2$FC < –1) (*Figure 3—figure supplement 1G*). Gene set enrichment analysis (GSEA) using the hallmark gene sets in the molecular signatures database (MSigDB) identified that pathways involved in oxidative phosphorylation and ROS were significantly downregulated in CBS-depleted AIS cells compared to the control AIS cells (*Figure 3G and H*). Therefore, altered mitochondrial energy metabolism and ROS production may contribute to CBS-dependent AIS maintenance.

## CBS mitochondrial localization is required for AIS maintenance

CBS has been reported to localize to both the cytoplasm and mitochondria and regulate mitochondrial function and ATP synthesis via $H_2S$ (*Bhattacharyya et al., 2013*; *Panagaki et al., 2019*). Consistent with this, we also observed CBS localization in the mitochondria by immunofluorescent cell staining (*Figure 4A and B*). AIS cells exhibited elevated mitochondrial abundance as indicated by increased intensity of MitoTracker-Deep Red staining compared to proliferating cells (*Figure 4A*), consistent with an increased abundance of proteins involved in the mitochondrial electron transport chain as detected by Western blotting (*Figure 4—figure supplement 1A*). The mitochondrial localization of CBS was further supported by Western blotting of mitochondrial extracts isolated from AIS and proliferating cells (*Figure 4C*). AIS cells displayed increased mitochondrial CBS abundance. To further validate CBS localization in mitochondria, we performed a protease protection assay using mitochondria isolated from cells expressing wild type CBS fused to a C-terminal FLAG tag (*Figure 4D*). The C-terminally FLAG-tagged CBS was present in intact mitochondria and was resistant to protease treatment and only degraded upon membrane solubilization by Triton X-100. A similar result was observed for the mitochondrial ATP synthase F1 subunit alpha ATP5A.

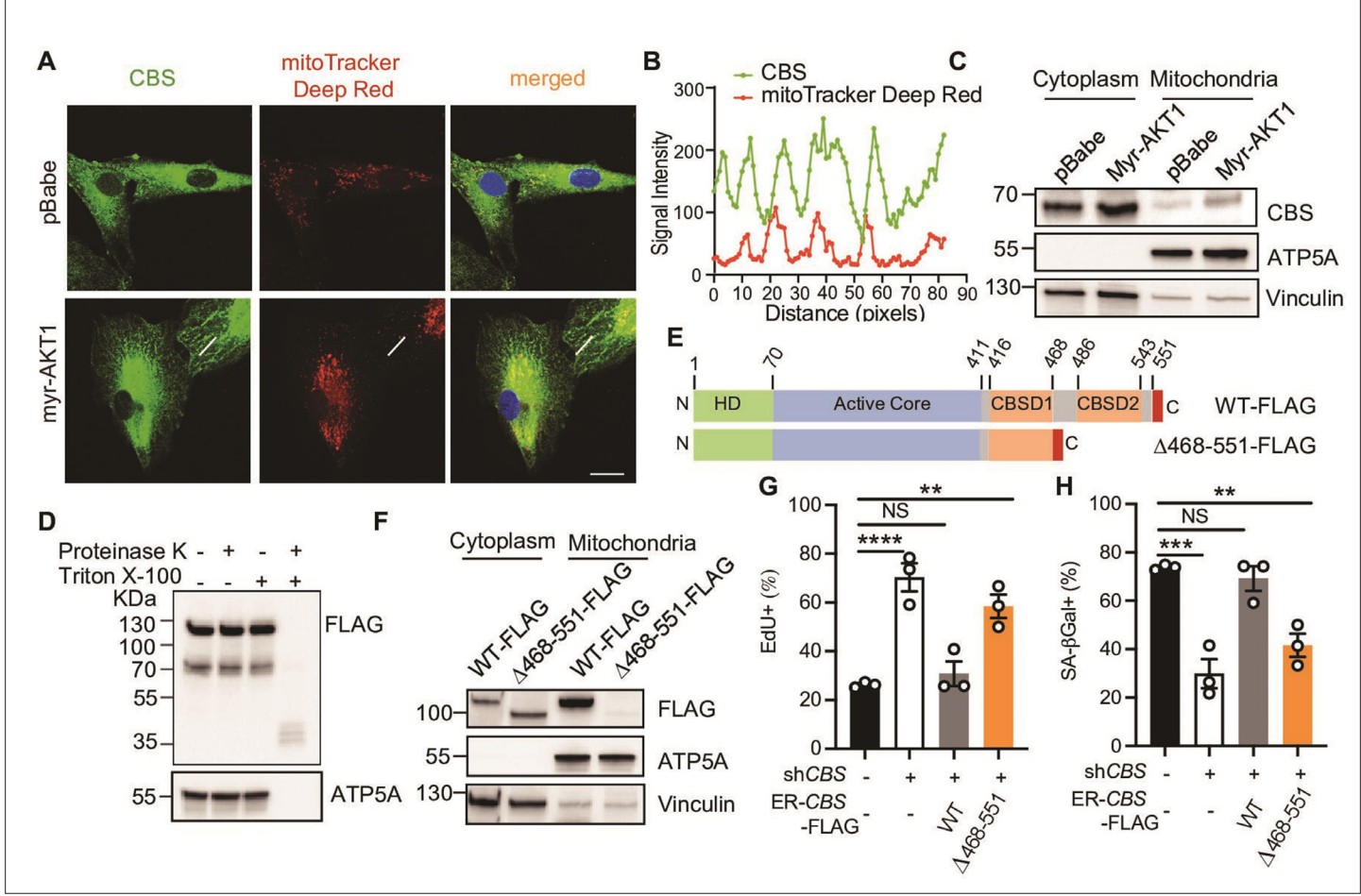

**Figure 4.** Cystathionine-β-synthase (CBS) mitochondrial localization is required for AKT-induced senescence (AIS) maintenance. (**A–B**) BJ3 human skin fibroblasts expressing telomerase reverse transcriptase (BJ-TERT) cells were transduced with pBabe or myrAKT1. Immunofluorescent staining showing CBS (green) and mitochondria (red) on day 10 post-transduction. The representative images are from one of two independent experiments. Scale bar = 20 μm. (**B**) Quantification of signal intensities using ImageJ by applying a single ROI to two color channels in the same image and extracting the plot profile. (**C**) Western blot analysis of CBS expression in the cytoplasmic and mitochondrial fractions isolated from BJ-TERT cells transduced with pBabe or myrAKT1. ATP5A and vinculin serve as the markers of mitochondria and cytoplasm, respectively. (**D**) Western blot analysis of a protease protection assay using the mitochondrial fraction isolated from BJ-TERT cells expressing C-terminal FLAG-tagged CBS. ATP5A serves as a positive control. (**E**) Schematic of 4-OHT-inducible plasmids expressing FLAG-tagged wild type CBS (WT) or a C-terminal regulatory domain CBSD2 truncated CBS mutant. (**F**) Western blot analysis of CBS expression in the cytoplasmic and mitochondrial fractions isolated from BJ-TERT cells transduced with FLAG-tagged wild type (WT) or a truncated mutant CBS after 20 nM 4-OHT induction for 3 days. ATP5A and vinculin serve as the markers of mitochondria and cytoplasm, respectively. (**C**), (**D**), and (**F**) are representative of at least n=3 experiments. (**G–H**) BJ-TERT cells expressing doxycycline-inducible *CBS* shRNA#2 and 4-OHT-inducible CBS wide type or a truncated mutant were transduced with myr-AKT1, treated with doxycycline (1 μg/ml)±4 OHT (20 nM) on day 5 post-transduction and analyzed on day 12 post-transduction. The percentage of cells with positive staining for (**G**) EdU proliferation marker incorporation or (**H**) SA-βGal activity is expressed as mean ± SEM (n=3). One-way analysis of variance (ANOVA) with Holm-Šídák's multiple comparisons was performed (NS, not significant; **p<0.01; ***p<0.001; ****p<0.0001).

The online version of this article includes the following source data and figure supplement(s) for figure 4:

**Source data 1.** Unedited immunoblots of *Figure 4C*.

**Source data 2.** Unedited immunoblots of *Figure 4D*.

**Source data 3.** Unedited immunoblots of *Figure 4F*.

**Figure supplement 1.** AKT-induced senescent cells showed increased expression of proteins involved in the mitochondrial electron transport chain.

**Figure supplement 1—source data 1.** Unedited immunoblots of *Figure 4—figure supplement 1A*.

Human CBS contains an N-terminal heme domain, catalytic core, and two CBS motifs (CBSD1 and CBSD2) in the C-terminal regulatory domain. A non-canonical mitochondrial targeting signal has been reported to reside within C-terminal CBSD2 motif (*Teng et al., 2013*). To confirm the CBSD2 motif is required for localizing CBS to mitochondria, BJ-TERT cells depleted of endogenous CBS was engineered to express wild type CBS or a C-terminal regulatory domain CBSD2 (Δ468–551) truncated CBS mutant (*Figure 4E*). Consistent with a previous finding (*Teng et al., 2013*), loss of CBSD2 motif abrogated CBS mitochondrial localization (*Figure 4F*). These results thus strongly support the mitochondrial localization of CBS through the CBSD2 motif.

To evaluate the functional significance of mitochondrial-localized CBS on AIS maintenance, we reconstituted CBS-depleted AIS-escaped cells with wild type or the CBSD2 truncation mutant. Expression of wild type CBS prevented AIS escape as evidenced by a decrease of EdU-positive cells and increase of SA-βGal-positive cells while cells expressing the truncation mutant still escaped from AIS (*Figure 4G and H*), demonstrating that mitochondrial localization of CBS is required to maintain AIS.

## CBS deficiency alleviates oxidative stress in AIS cells

To investigate the role of CBS-mediated mitochondrial alterations in AIS maintenance, we examined the oxidative phosphorylation status in AIS cells transfected with control or *CBS* siRNA using a Seahorse extracellular flux analyzer (*Figure 5A*). The oxygen consumption rate (OCR) and ATP production were markedly elevated during AIS and knockdown of *CBS* significantly suppressed basal OCR and ATP production (*Figure 5B and C*). These results suggest that CBS is required for activated oxidative phosphorylation during AIS maintenance and CBS depletion reduces mitochondrial bioenergetics in AIS. To test whether these effects were specific for AIS, we also performed the Seahorse analysis on cells during RIS. While basal OCR was also increased during RIS, knockdown of *CBS* in RIS cells, in contrast to those undergoing AIS, had an opposite effect on oxidative phosphorylation by further upregulating basal OCR and ATP production (*Figure 5B and C*). This distinct effect on mitochondrial activity is likely to contribute to the specific regulatory role of CBS in AIS maintenance. To further confirm that CBS-stimulated oxidative phosphorylation of AIS cells relies on mitochondrial localization, we performed the Seahorse analysis in AIS cells expressing either wild type or the CBS truncation mutant (*Figure 5D* and *Figure 5—figure supplement 1A*). Reconstitution with wild type CBS rescued basal OCR and ATP production levels in CBS-depleted AIS cells. In contrast, AIS cells expressing C-terminal truncated CBS protein failed to restore basal OCR and ATP production (*Figure 5E and F*). These mitochondrial alterations corresponded to the change of senescence status observed in *Figure 5G and H*. Collectively, our results strongly support the concept that AKT overexpression promotes CBS translocation to mitochondria and subsequently increases energy metabolism to sustain the senescence state.

Mitochondria are the major intracellular organelles of ROS production. Elevated ROS results in oxidative stress which may underlie AIS. To test this hypothesis, we first treated AIS cells with an antioxidant Trolox, which resulted in increased proliferation (*Figure 4G*) and decreased SA-βGal staining (*Figure 5—figure supplement 1B*), establishing the role of oxidative stress in AIS maintenance. To test the impact of AKT hyperactivation on mitochondrial and cytoplasmic ROS production, we performed flow cytometry analysis using MitoSOX and $H_2$DCFDH-DA, respectively, on proliferating and AIS cells transfected with control or *CBS* siRNA. Critically, *CBS* knockdown decreased ROS levels in AIS cells (*Figure 5H and I* and *Figure 5—figure supplement 1C*).

Together, these results strongly support the concept that increased oxidative phosphorylation and ROS production sustain AIS status that require CBS. In parallel, AKT activation increases exogenous cysteine import and transsulfuration pathway activity, which consequently stimulates GSH and $H_2S$ production, thereby protecting AIS cells from ROS-induced cell death. Importantly, this AKT-dependent increase of the antioxidant capacity is retained in CBS-deficient cells and thus contributes to escape of AIS cells from cell cycle arrest (*Figure 4J*).

## CBS expression is frequently suppressed in gastric cancer

Given that we showed CBS loss promotes escape from AIS, we hypothesized that loss of CBS could cooperate with oncogenic activation of the PI3K/AKT/mTORC1 pathway to promote tumorigenesis. Analysis of TCGA stomach adenocarcinoma data from 478 samples using cBioPortal (http://www.cbioportal.org) identified *CBS* deep deletions and mutations in gastric cancer (*Figure 6A*). Further

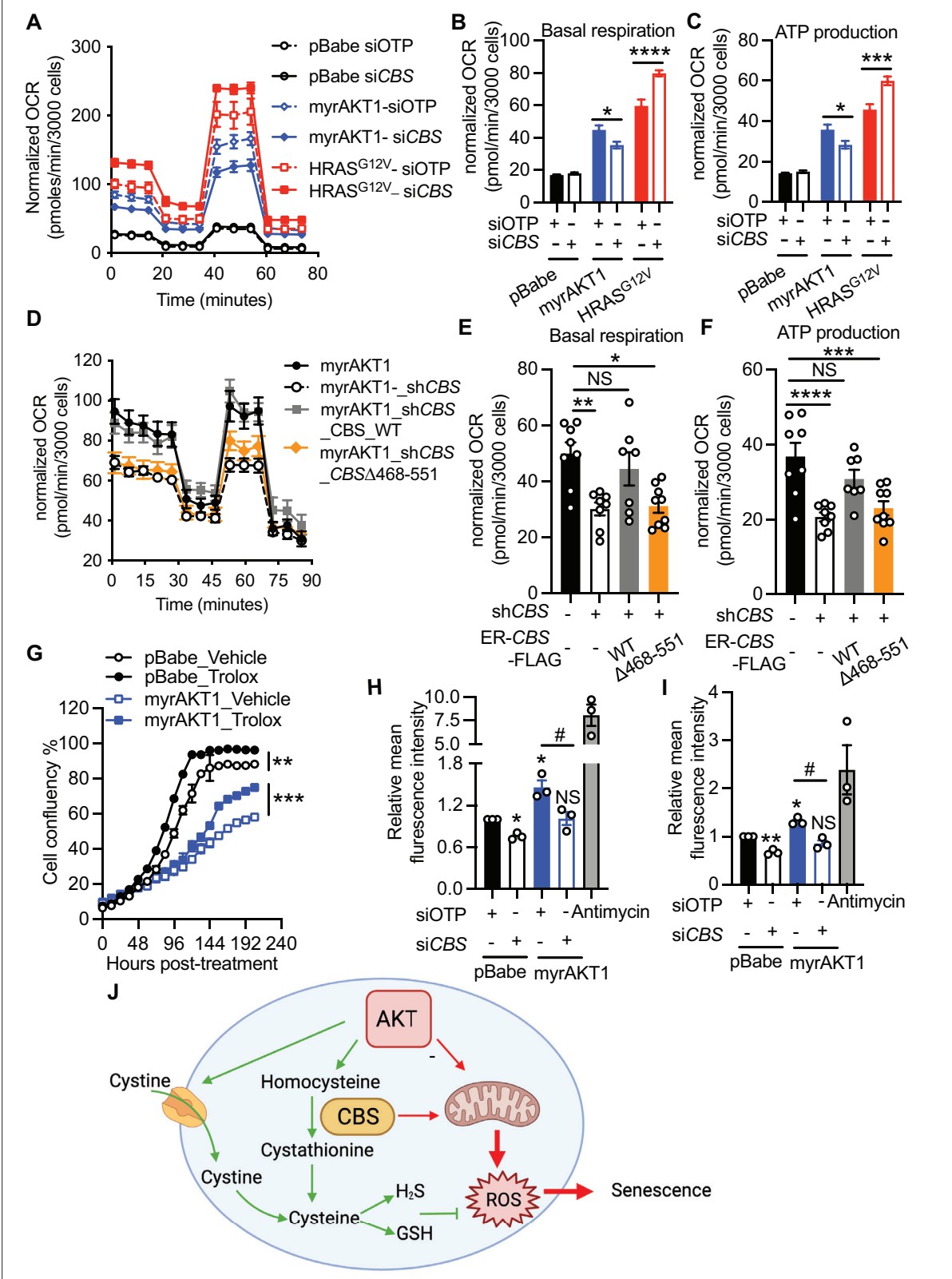

**Figure 5.** Cystathionine-β-synthase (CBS) deficiency alleviates oxidative stress in AKT-induced senescence (AIS) cells. (**A–C**) BJ3 human skin fibroblasts expressing telomerase reverse transcriptase (BJ-TERT) cells were transduced with either pBabe, myrAKT1, or HRAS[G12V]. After 5 days cells were transfected with either *CBS* siRNA (siCBS) or control siRNA (siOTP) and analyzed on day 6 post-siRNA transfection. (**A**) Oxygen consumption rate (OCR) was determined using the Seahorse XF96 mitochondrial stress test by sequential injection of oligomycin (1 µM), *p*-trifluoromethoxy- phenylhydrazone

*Figure 5 continued on next page*

*Figure 5 continued*

(FCCP) (1 µM), and rotenone/antimycin (0.5 µM each). (**B**) Basal respiration rate and (**C**) ATP production rate determined by the mitochondrial stress test are presented as mean ± SEM of four replicates from one of two independent experiments. Two-way analysis of variance (ANOVA) with Holm-Šídák's multiple comparisons was performed (*p<0.05; ***p<0.001; ****p<0.0001). (**D–F**) BJ-TERT cells expressing doxycycline-inducible *CBS* shRNA#2 and 4-OHT-inducible CBS wide type or a truncated mutant were transduced with myrAKT1, treated with ±doxycycline (1 µg/ml)±4 OHT (20 nM) on day 5 post-transduction and analyzed on day 12 post-transduction. (**D**) OCR, (**E**) basal respiration rate, and (**F**) ATP production rate determined by the Seahorse XF96 mitochondrial stress test were presented as mean ± SEM (n=7–9 from one of two independent experiments). One-way ANOVA with Holm-Šídák's multiple comparisons was performed (NS, not significant; *p<0.05; **p<0.01; ***p<0.001; ****p<0.0001). (**G**) On day 5 post-transduction, cells were treated with Trolox 100 µM. Cell confluency measured by IncuCyte was presented as mean ± SEM (n=3). Statistical significance at the last time point was determined by unpaired student's t-test (**p<0.01; ***p<0.001). (**H and I**) Flow cytometric analysis of (**H**) the mitochondrial superoxide production by MitoSOX and (**I**) the cytoplasmic reactive oxygen species (ROS) production by H$_2$DCFDH-DA on day 6 post-siRNA transfection. Relative mean fluorescence intensity normalized to the pBabe-siOTP is presented as mean ± SEM (n=3). Statistical significance was determined by one sample t-test compared to the hypothetical value 1.0 (NS, not significant; *p<0.05; **p<0.01) and one-way ANOVA with Holm-Šídák's multiple comparisons (#p<0.05). (**J**) Schematic diagram illustrating CBS-mediated metabolic alterations on maintenance of AKT-induced senescence.

The online version of this article includes the following source data and figure supplement(s) for figure 5:

**Figure supplement 1.** Cystathionine-β-synthase (CBS) deficiency alleviates oxidative stress in AKT-induced senescence (AIS) cells.

**Figure supplement 1—source data 1.** Unedited immunoblots of *Figure 5—figure supplement 1A*.

analysis of the TCGA gastric cancer patient data revealed that PI3K/AKT/mTORC1 signaling pathway alterations occur in 33% of gastric cancer (142 of 441 samples) with *PIK3CA*, *AKT,* and *PTEN* alterations being the most common genetic alterations (*Figure 6—figure supplement 1A*). *CBS* genetic alterations were found in 18 of 441 gastric cancers with 12 cases cooccurring with mutations in PI3K/AKT/mTORC1 pathway. There was a significant association between *CBS* mutations and PI3K/AKT/mTORC1 pathway activation (p-value = 0.0031, 95% CI = 1.5202–14.9139 and odds ratio 4.4905). Furthermore, *CBS* mRNA level was significantly decreased in tumors (N=406) compared to adjacent normal tissues (N=211) in gastric cancer patients (*Figure 6B*), consistent with the hypothesis that CBS is a tumor suppressor in gastric cancers characterized by PI3K/AKT/mTORC1 pathway alterations.

To evaluate alteration of CBS protein expression in human gastric cancer, we assessed CBS protein levels in paired samples of gastric tumors and adjacent non-cancerous mucosa from 62 gastric cancer patients in a tissue microarray (TMA) using immunofluorescent staining (*Figure 6C*). This TMA was assembled from paraffin-embedded tissue blocks collected from gastric cancer patients who underwent gastrectomy from 2000 to 2005 in Changhai Hospital, Shanghai, China, as described previously (*Zhang et al., 2013*). Cytosolic CBS protein expression level, as measured by the percentage of cells with positive CBS staining (*Figure 5D*), and fluorescence intensity (*Figure 6E*) were significantly downregulated in tumor tissues compared to the adjacent normal gastric tissues.

To establish a cell-based system to probe the interaction between activated PI3K/AKT/mTORC1 signaling and loss of CBS expression, we first examined CBS protein expression in six gastric cancer cell lines compared with an SV40-transformed gastric epithelial cell line GES-1, which was derived from fetal stomach mucosa and was non-tumorigenic in nude mice (*Ke et al., 1994*). Compared to gastric epithelial cells, CBS expression was markedly decreased in all gastric cancer cell lines while elevated AKT activity, as indicated by increase of AKT phosphorylation, was observed in AGS, Hs746T, KATO III gastric cancer cell lines (*Figure 6F*).

Previous studies have demonstrated *CBS* deficiency in human gastric cancer cells with hypermethylation at the *CBS* promoter region, implicating loss of CBS through epigenetic regulation in gastric cancer (*Zhao et al., 2012*). Comparison of *CBS* mRNA expression with DNA methylation status in a panel of 34 human gastric cancer cell lines revealed a significant negative correlation between *CBS* mRNA expression level and DNA methylation (*Figure 6G*). Indeed, the *CBS* promoter was methylated in all six gastric cancer cell lines tested, and half of them displayed complete absence of unmethylated *CBS* promoter (SNU-1, NCI-N87, and AGS) that was associated with undetectable *CBS* transcription (*Figure 6—figure supplement 1D*). Blocking DNA methylation with azacitidine, a DNA methyltransferase-inhibiting cytosine nucleoside analogue upregulated *CBS* mRNA expression in all cell lines tested except Hs746T, whereby SNU-1 cells showed a>50-fold increase, confirming epigenetic silencing of *CBS* expression in gastric cancer cells (*Figure 6—figure supplement 1E*). Given the GES-1 cells express high levels of CBS as well as activated AKT signaling comparable to

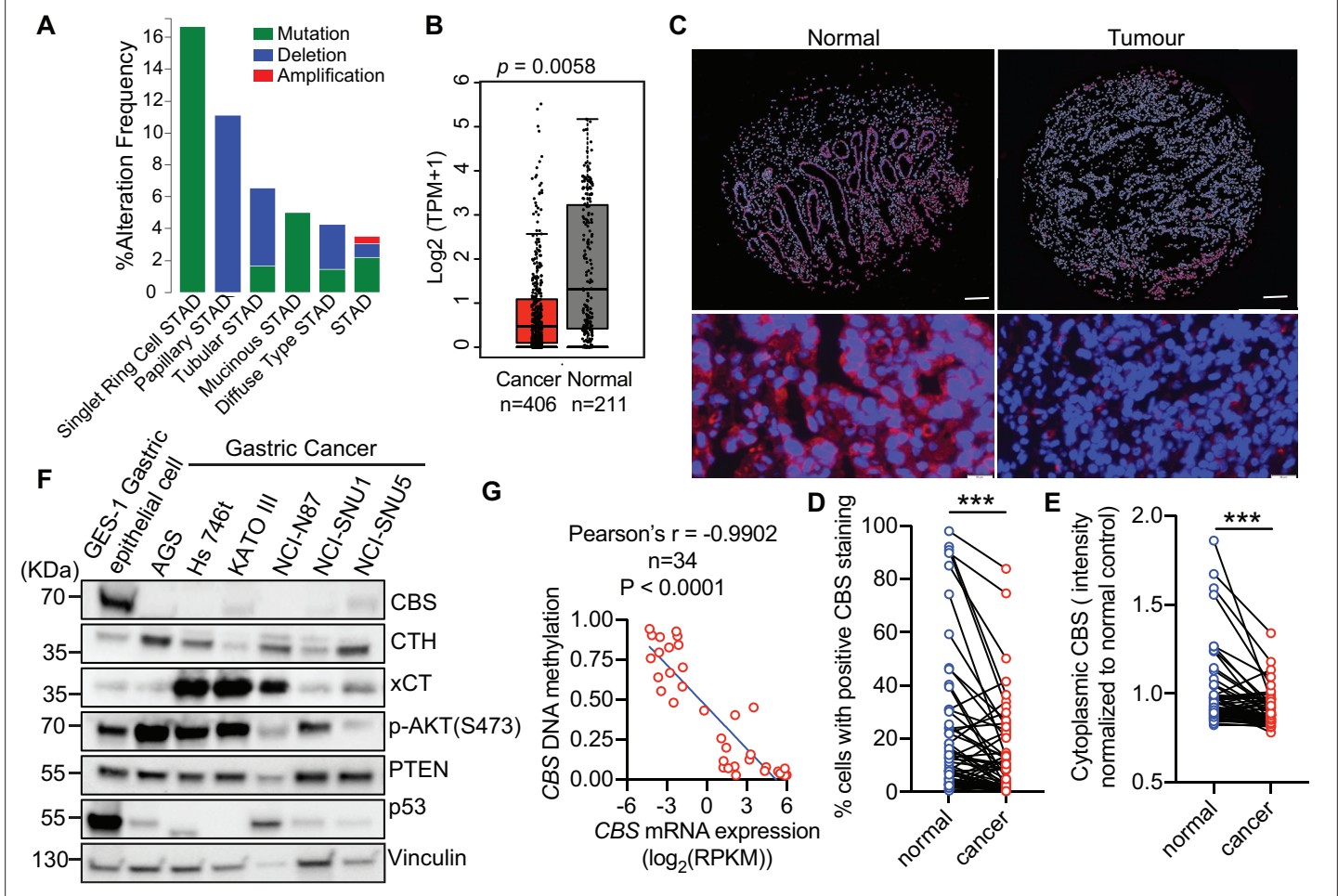

**Figure 6.** Cystathionine-β-synthase (CBS) expression is suppressed in tumor tissues and human cell lines of gastric cancer. (**A**) Analysis of genetic alterations of *CBS* in different subtypes of gastric cancer using cBioPortal (http://www.cbioportal.org). (**B**) Median mRNA expression level of *CBS* in normal stomach tissues and stomach adenocarcinoma tissues profiled by Gene Expression Profiling Interactive Analysis (GEPIA, http://gepia.cancer-pku.cn) based on the TCGA database. p=0.0058 by unpaired Student's t-test. (**C–E**) Gastric cancer patient tissue microarray was assessed by immunofluorescent staining of CBS (red) and counterstained for the nucleus (DAPI, blue). (**C**) The representative images of normal or cancer tissues from one gastric cancer patient were shown. Top panel: Scale bars = 200 μm. Bottom panel: Scale bars = 50 μm. (**D**) The percentage of cytosolic CBS-positive cells and (**E**) the intensity of cytoplasmic CBS in the tumor or adjacent normal tissues from 62 patients were shown. Statistical significance was determined by paired Student's t-test (***p<0.001). (**F**) Western blot analysis in GES-1 gastric epithelial cell line and six gastric cancer cell lines. Vinculin was probed as a loading control. Representative of at least n=3 experiments. (**G**) Correlation of *CBS* mRNA expression with *CBS* DNA methylation in 34 gastric cancer cell lines based on the data retrieved from the Cancer Cell Line Encyclopedia.

The online version of this article includes the following source data and figure supplement(s) for figure 6:

**Source data 1.** Unedited immunoblots of *Figure 6F*.

**Figure supplement 1.** Cystathionine-β-synthase (CBS) expression is suppressed in tumor tissues and human cell lines of gastric cancer.

**Figure supplement 1—source data 1.** Unedited gel image of *Figure 6—figure supplement 1D*.

some of gastric cancer cell lines, we used this cell line to interrogate the role of CBS loss in the initiation of gastric cancer.

## Loss of CBS cooperates with PI3K/AKT pathway activation to promote gastric cancer pathogenesis

To further test if CBS loss cooperates with PI3K/AKT/mTORC1 hyperactivation in gastric cancer oncogenesis, we transduced myrAKT1 and *CBS* shRNA into GES-1 cells and tested their ability to form colonies in soft agar (*Figure 7A*). Consistent with our results in fibroblasts, AKT hyperactivation increased transsulfuration pathway activity and GSH production, which was not affected by CBS depletion

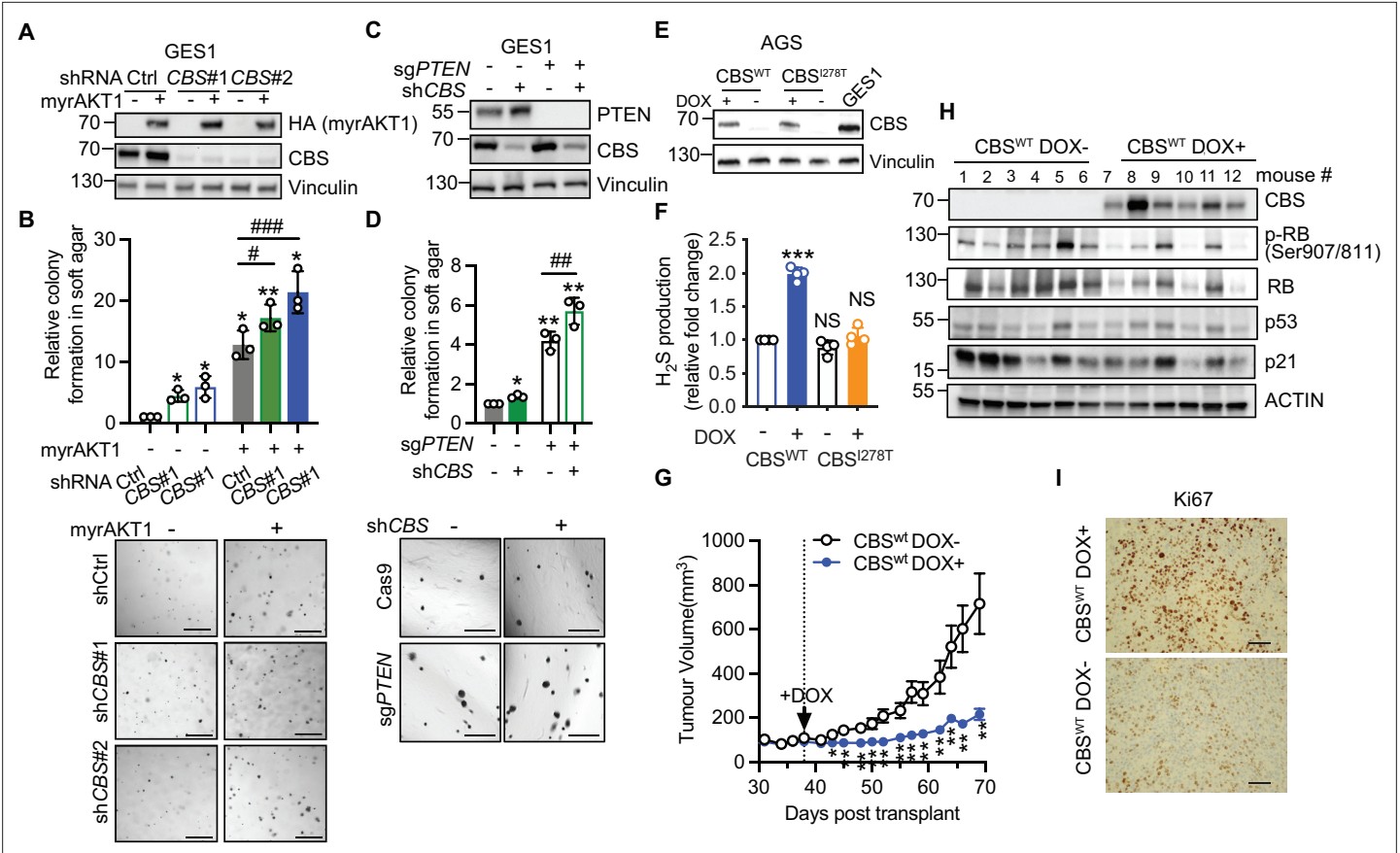

**Figure 7.** Loss of cystathionine-β-synthase (CBS) synergizes with PI3K/AKT pathway to promote gastric cancer pathogenesis. (**A–B**) GES-1 gastric epithelial cells were stably transfected with doxycycline-inducible myrAKT1 and pGIPZ-sh*CBS* or non-silencing small hairpin RNA (shRNA) control (shCtrl). (**A**) Western blot analysis showing HA-tagged myrAKT1 and CBS after doxycycline (0.75 μg/ml) induction for 3 days. Vinculin was probed as a loading control. (**B**) Relative colony formation assessed by soft agar colony formation assay on day 21 post-doxycycline induction was normalized to doxycycline-untreated shCtrl group and presented as mean ± SEM (n=3). The representative images of colonies formed in the soft agar. Scale bar = 500 μm. (**C–D**) GES-1 cells with *PTEN* knockout by CRISPR or Cas9 control were transduced with doxycycline-inducible *CBS* shRNA#1. (**C**) Western blotting after treatment with doxycycline (0.75 μg/ml) for 3 days. Vinculin was probed as a loading control. (**D**) Relative colony formation assessed by soft agar colony formation assay on day 21 post-doxycycline induction was normalized to doxycycline-untreated shCtrl group and presented as mean ± SEM (n=3). The representative images of colonies formed in the soft agar. Scale bar = 500 μm. (**E–F**) AGS gastric cancer cells were stably transfected with doxycycline-inducible CBS^wt or CBS^I278T. Cells were cultured in the presence or absence of doxycycline induction (0.08 μg/ml for CBS^wt or 1 μg/ml for CBS^I278T). (**E**) Western blot analysis on day 3 post-treatment. Vinculin was probed as a loading control. (**F**) Fold changes of hydrogen sulfide ($H_2S$) production over doxycycline-untreated CBS^wt group were presented as mean ± SEM (n=4). Statistical significance in (**A, C, and F**) was determined by one-sample t-test compared to the hypothetical value 1.0 (NS, not significant; *p<0.05; **p<0.01; ***p<0.001) and two-way analysis of variance (ANOVA) with Holm-Šídák's multiple comparisons (#p<0.05; ##p<0.01; ###p<0.001). (**G–I**) AGS gastric cancer cells expressing doxycycline-inducible CBS^wt were subcutaneously implanted in BALB/c nude mice. On day 38 post-implantation, mice were supplied with water containing 0.2% (W/V) doxycycline and 600 mg doxycycline/kg food. (**G**) Tumor volume was presented as mean ± SEM (n=6 mice per group). Statistical significance at each time point was determined by unpaired Student's test (*p<0.05; **p<0.01). (**H**) Western blot analysis of CBS and senescence-associated protein expression in tumor tissues. Actin was probed as a loading control. (**I**) IHC of Ki67 on the tumor tissue. Scale bar: 50 μm.

The online version of this article includes the following source data and figure supplement(s) for figure 7:

**Source data 1.** Unedited immunoblots of *Figure 7A*.

**Source data 2.** Unedited immunoblots of *Figure 7C*.

**Source data 3.** Unedited immunoblots of *Figure 7E*.

**Figure supplement 1.** Loss of cystathionine-β-synthase (CBS) synergizes with the PI3K/AKT pathway to promote gastric cancer pathogenesis.

(*Figure 7—figure supplement 1A, B*). Hyperactivation of AKT1 had no significant effect on cell proliferation in 2D culture and did not induce senescence due to inactivation of p53 and pRb by SV40T in the GES-1 cells (*Figure 6—figure supplement 1C*). However, it effectively promoted anchorage-independent growth, and this was further enhanced by CBS depletion (*Figure 7B*). CBS depletion also significantly enhanced anchorage-independent growth of GES-1 cells with clustered regularly interspaced short palindromic repeats (CRISPR)/Cas9-mediated *PTEN* knockout (*Figure 7C–D*) but not in 2D cell culture (*Figure 7—figure supplement 1D*). These results support that CBS loss can cooperate with PI3K/AKT signaling to promote oncogenic transformation.

To further investigate the functional cooperation of CBS and PI3K/AKT signaling in gastric cancer pathogenesis, we engineered AGS gastric cancer cells, which harbor *CBS* deficiency and *PIK3CA* mutations E545A and E453K resulting in AKT activation (*Figure 6F*), to express a doxycycline-inducible wild type CBS or an inactive CBS$^{I278T}$ mutant. This mutation is the most frequently observed CBS mutation in cancer cells and exhibits only ~2.4% of the enzyme activity of wild type CBS (*Kruger and Cox, 1995*). Treatment of AGS cells with doxycycline restored CBS$^{wt}$ and CBS$^{I278T}$ protein expression to a level comparable to that of GES-1 gastric epithelial cells (*Figure 7E*). Restoration of wild type CBS increased H$_2$S production (*Figure 7F*) but did not significantly enhance GSH abundance (*Figure 7—figure supplement 1E*). Interestingly, restoration of CBS expression did not affect AGS cell proliferation in 2D cell culture (*Figure 7—figure supplement 1F*).

To evaluate the functional impact of CBS restoration in vivo, we transplanted the AGS cells expressing doxycycline-inducible CBS into immunocompromised mice. Induction of CBS$^{wt}$ significantly suppressed AGS tumor growth (*Figure 7G*). Induction of CBS was also associated with a marked decrease in Ki67 expression and inhibitory RB phosphorylation without altering p53 and p21 expression levels in the tumor tissues, suggesting that restoration of CBS expression could suppress gastric tumor formation independent of p53 (*Figure 7H, I*).

## Discussion

Hyperactivation of the PI3K/AKT/mTORC1 signaling pathway causes a senescence-like phenotype in non-transformed cells, which acts as a protective brake against tumor formation (*Zhu et al., 2020*). Subsequent genetic or epigenetic changes can disengage this brake and lead to oncogenic transformation. Deregulated metabolism along with cell cycle withdrawal, SASP, and macromolecular damage are hallmarks of the senescence phenotype (*Gorgoulis et al., 2019*). Oxidative stress is a key metabolic feature of RAS-induced senescence and ROS triggers DNA damage and proliferative arrest in RIS cells (*Irani et al., 1997*; *Lee et al., 1999*; *Ogrunc et al., 2014*). Activation of AKT can also increase intracellular ROS levels by stimulating oxidative phosphorylation and impairing ROS scavenging by inhibition of FoxO transcription factors (*Nogueira et al., 2008*). Consistent with these findings, we demonstrate increased mitochondrial abundance and respiratory activity, as well as ROS production in the senescent-like cells resulting from AKT hyperactivation. Alleviation of oxidative stress by antioxidant treatment partially releases AIS cells from cell division arrest, supporting that ROS is required for AIS maintenance.

On the other hand, activation of the PI3K/AKT pathway has been observed to induce a potent antioxidant response (*Hoxhaj and Manning, 2020*) that may antagonize the tumor-suppressive AIS. One major ROS-scavenging mechanism by the PI3K/AKT pathway is through sustained activation of nuclear factor erythroid 2-related factor 2 (NRF2) (*Mitsuishi et al., 2012*; *Rada et al., 2011*). In this study we uncovered another mechanism of AKT-mediated ROS detoxification occurring through increased cysteine import and enhanced GSH and H$_2$S synthesis (*Figure 5J*). We found that AKT activation markedly increased xCT protein expression, a cystine-glutamate antiporter encoded by *SLC7A11*, indicating the increase of extracellular cystine uptake in AIS. Isotope tracing coupled with LC-MS analysis revealed that in cysteine-replete conditions AIS cells exhibited an elevated intracellular cysteine level and subsequent increased production of GSH and H$_2$S, the critical components in the antioxidant system, which was abrogated by cysteine deprivation. Previous studies have shown that H$_2$S inhibits H$_2$O$_2$-mediated mitochondrial dysfunction by preserving the protein expression levels and activity of key antioxidant enzymes, inhibiting ROS production and lipid peroxidation (*Wen et al., 2013*). Moreover, these effects may be associated with sulfhydration of Keap1 and activation of Nrf2 or increased production of the antioxidant GSH (*Koike et al., 2013*; *Yang et al., 2013*). In addition to antioxidant effects, H$_2$S can modulate mitochondrial functions and cellular bioenergetics

in a concentration-dependent manner. At low concentrations, $H_2S$ acts as a mitochondrial electron donor to mitochondrial complex II, resulting in bioenergetic stimulation (*Szabo et al., 2013*). At high concentrations, $H_2S$ acts as a mitochondrial poison via the inhibition of cytochrome *c* oxidase in mitochondrial complex IV (*Panagaki et al., 2019*; *Szabo et al., 2014*). Our finding that suppression of $H_2S$ by AOAA exacerbates proliferative arrest of AKT-hyperactivated cells supports a protective effect of $H_2S$ from oxidative stress-induced cell death (*Figure 1D*). The protective role of the transsulfuration pathway in AIS is further supported by the finding that myrAKT1-expressing cells are resistant to exogenous cysteine deprivation (*Figure 1E*). Thus, we propose that increased levels of GSH and $H_2S$ through the transsulfuration pathway during AIS maintenance enhance the antioxidant capacity of AIS cells, protecting senescent cells from ROS-induced cell death (*Figure 5J*).

Paradoxically, depletion of CBS, a key enzyme involved in the transsulfuration pathway, led to AIS escape, indicating that CBS is required for AIS maintenance. Indeed, we found that under cysteine-replete conditions, CBS depletion did not affect the production of antioxidants (GSH and $H_2S$) downstream of the transsulfuration pathway. Instead, in AIS cells CBS mitochondrial localization was enhanced, resulting in an increase of ROS production through upregulated mitochondrial oxidative phosphorylation, which contributes to maintenance of AIS status (*Figure 5J*). Intriguingly, a recent publication reported that overproduction of $H_2S$ by increased mitochondrial-localized CBS expression results in suppression of mitochondrial oxidative phosphorylation and ATP production in the fibroblasts from Down syndrome patients (*Panagaki et al., 2019*). This discrepancy could be partially explained by the bell-shaped or biphasic biological effect of $H_2S$ as described above.

Oncogene-induced senescence acts as a critical tumor-suppressive brake and this senescence brake is disengaged during tumorigenesis. Based on our observation that loss of CBS promoted AIS escape in normal cells, we propose a potential tumor-suppressive role for CBS in cancers harboring PI3K/AKT pathway activation. We demonstrated suppression of *CBS* expression in primary gastric tumor tissues and a panel of human gastric cancer cell lines through epigenetic silencing. In parallel, a negative correlation between CBS and xCT expression was observed in gastric cancer cell lines, which is consistent with a recent report that cancer cells maintain intracellular cysteine levels and sustain cell proliferation by increasing xCT-mediated cysteine uptake (*Zhu et al., 2019*). We further demonstrated that loss of CBS cooperates with AKT hyperactivation to promote anchorage-independent growth of gastric epithelial cells and restoration of CBS expression inhibited PI3K/AKT hyperactive gastric tumor growth. Induction of apoptosis and impairment of gastric cancer cell metastasis by increasing $H_2S$ production upon NaHS treatment has been previously reported (*Zhang et al., 2015*). Whether the tumor-suppressive effect of CBS in gastric cancer cells is mediated through $H_2S$ requires further investigation.

Taken together, our study identifies CBS as a novel regulator of AIS maintenance and a potential tumor suppressor in gastric cancer pathogenesis, potentially providing a new metabolic vulnerability that can be harnessed to target PI3K/AKT/mTORC1-driven cancers.

## Materials and methods
### Cell culture and reagents

BJ-TERT-immortalized human foreskin fibroblasts were a gift from Robert Weinberg (Massachusetts Institute of Technology, Cambridge, MA). Primary IMR-90 lung fibroblasts originating from the American Type Culture Centre (ATCC, ATCC-CL-186) were obtained from the Garvan Institute of Medical Research, Sydney, Australia. Human embryonic kidney (HEK293T) cells were purchased from the ATCC (ATCC-CRL-3216). Human gastric cancer cell lines, AGS (ATCC-CRL-1739), Hs 746T (ATCC-HTB-135), KATO III (ATCC-HTB-103), NCI-N87 (ATCC-CRL-5822), SNU1 (ATCC-CRL-5971), and SNU5 (ATCC-CRL-5973) were obtained from the ATCC. The human fetal gastric epithelial cell line GES-1 was provided by Prof. Caiyun Fu (Zhejiang Sci-Tech University, China). These cell lines were authenticated by STR profiling and tested for mycoplasma contamination prior to experimentation and intermittently tested thereafter by PCR. BJ-TERT cells were cultured in Dulbecco's modified Eagle's medium (DMEM) plus 20 mM HEPES, 17% Medium 199 (Gibco #11150067), 15% fetal bovine serum (FBS), and 1% GlutaMAX (Gibco #35050061). IMR90 cells were cultured in Eagle's minimum essential medium (EMEM) supplemented with 10% FBS, 5 mM sodium pyruvate (Gibco, #11360070), 1% non-essential amino acids (Gibco, #11140050), and 1% GlutaMAX. GES-1, AGS, Hs746T, and KATOIII were

cultured in DMEM+20 mM HEPES, 10% FBS, and 1% GlutaMAX. NCI-N87 and SNU-1 were cultured in RPMI+20 mM HEPES, 10% FBS, and 1% GlutaMAX. SNU-5 was cultured in IMDM, 20%FBS, and 1% GlutaMAX.

## Plasmids

The plasmid pBabe-puro was a gift from Morgenstern, Land, and Weinberg (Addgene plasmid #1764) (*Morgenstern and Land, 1990*), pBabe-puro-myr-AKT1 and pBabe-puro-HRAS$^{G12V}$ were described previously (*Astle et al., 2012*). HA-myrAKT1 was directly subcloned into pCW57.1 (Addgene plasmid #41393) to generate doxycycline-inducible pCW57.1-myrAKT1. The REBIR construct, a modified doxycycline-inducible mirE shRNA expression vector, was a gift from Dr Sang-Kyu Kim (*Kim et al., 2018*). The Dharmacon GIPZ lentiviral shRNAs targeting human *CBS* gene were obtained from Horizon Discovery, UK, and the knockdown efficacy was tested in BJ-TERT cells. The two best shRNAs were subcloned into the REBIR construct. 97-mer shRNA sequences are listed in Appendix 2.

The cloning vector pBSK(+) containing the complementary DNA (cDNA) encoding human CBS isoform 1 was synthesized and purchase from Biomatik Corporation, Canada. *CBS* cDNA was subcloned into REBIR plasmid in which the dsRed2/mirE cassette has been removed and eBFP2 was replaced with the puromycin resistance gene. The resulting plasmid was designated pRT3-puro-CBS. pLNCX2 ER:ras was a gift from Masashi Narita (Addgene plasmid #67844) (*Young et al., 2009*). FLAG-tagged wild type CBS or a mutant with a deletion of the C-terminal regulatory domain CBSD2 (Δ468–551) was subcloned into this vector to generate pLNCX2-CBS WT-FLAG and pLNCX2-CBS Δ468–551-FLAG plasmids.

## Virus production and transduction

HEK293T cells were seeded 24 hr prior to transfection in tissue culture flasks at 80–90% confluency. To generate the retrovirus, the transfection reagent master mixes were prepared by combining equal mass of plasmid DNA vectors, pEQ-PAM3-E, and RD114 envelope plasmid as previously described (*Gavrilescu and Van Etten, 2007*). To generate the lentivirus, plasmid DNA vectors were combined with pMDL, pRSV-REV, and pCMV-VSV-G packaging plasmids at the ratio of mass 3:1:1:1. The plasmid mixtures were combined with polyethylenimine (PEI, 5 μg per μg plasmid), a cationic polymer for gene delivery. After vortex briefly, the mixtures were incubated at room temperature (RT) for 25 min and then added to the media in a dropwise manner. The culture media were refreshed at 24 hr post-transfection. At 48 and 54 hr post-transfection, the culture media containing viral particles was collected and passed through a 0.4 μm filter. Virus-containing media were concentrated and stored at –80°C for later use.

Cells were seeded 24 hr before virus infection at 60–70% confluency. The concentrated virus was added with 4 μg/ml polybrene. At 48 hr post-transduction, the media was removed and replaced with the complete media. The transduced cells were selected either by the defined antibiotics or by cell sorting for the fluorescent marker.

## siRNA transfection of BJ-TERT and IMR-90 fibroblasts

The siRNA targeting *CBS* (#L-008617-00) and On-Target Plus Non-Targeting siRNA control (#D-001810–01) were purchased from Horizon Discovery, UK. siRNA transfection was performed by reverse transfection where cells were seeded onto plates containing transfection reagents and siRNA mixture. In 96-well plates, 0.1 μl (For BJ-TERT) or 0.4 μl (For IMR-90) Dharmafect 1 (Dharmacon-T-2001-01) and 20 nM ON-TARGETplus SMARTPool siRNA per well were mixed and dispensed into each well. 4000 BJ-TERT or 12,000 IMR-90 cells in a total volume of 100 μl per well was then added. The negative controls including cell suspension only (no Dharmafect, no siRNA) or Dharmafect only control (no siRNA) were also prepared. The fresh complete medium was replaced at 24 hr post-transfection.

## CRISPR/Cas-9 gene deletion of *PTEN*

The *PTEN* sgRNA sequence (TCATCTGGATTATAGACCAG) was cloned into FgH1t-puro plasmid. Cells were infected at 0.3 MOI with Cas9-expressing lentivirus and then sorted for mCherry fluorescence marker. Cells were then subsequently re-infected with lentiviruses expressing *PTEN* sgRNA and selected by puromycin.

## Mitochondrial and cytoplasmic fractionation and protease protection assay

The mitochondrial and cytoplasmic fractions were isolated using the Qproteome mitochondria isolation kit (Qiagen #37612) according to the manufacturer's protocol; 700,000 cells were seeded per 15 cm plate and five plates per cell line were prepared. Cells were cultured for 3 days prior to harvesting for fractionation.

The protease protection assay was performed as described previously with modifications (*Mani et al., 2017*); 30–50 µg isotonically isolated mitochondrial was resuspended in 20 mM Tris-HCl pH 7.2, 15 mM $KH_2PO_4$, 20 mM $MgSO_4$, 0.6 M sorbitol in 50 µl in the presence and the absence of proteinase K (0.1 µg/ml), and 0.5% Triton-X100 and incubate on ice for 3 min followed by centrifugation at 8000 *g* at 4°C for 5 min. The pellets were resuspended in SDS lysis buffer (0.5 mM EDTA, 20 mM HEPES, 2% (w/v) SDS pH 7.9) and subjected to Western blotting analysis.

## Western blotting analysis

Protein was extracted with SDS-lysis buffer (0.5 mM EDTA, 20 mM HEPES, 2% (w/v) SDS pH 7.9) and the protein concentrations were determined with the Bio-Rad DC protein assay. Proteins were resolved by SDS-PAGE, transferred to PVDF membranes, and immunoblotted with primary and horseradish peroxidase-conjugated secondary antibodies (Appendix 1—Key resources table). The signals were visualized by Western-Lightning Plus ECL (Perkin-Elmer-NEL104001EA) and ChemiDoc Imaging system (Bio-Rad-17001401).

## Gene expression analysis by quantitative real-time PCR

RNA isolation and purification were performed using the ISOLATE-II kit (Bioline #52073) according to the manufacturer's protocol; 500 ng of purified RNA was treated with DNase at 37°C for 15 min followed by heat inactivation at 70°C for 15 min. cDNA synthesis by reverse transcription was performed using SuperScript III First-Strand Synthesis System per manufacturer's instruction under the following conditions: initial incubation at 37°C for 5 min, reverse transcription by SuperScript III reverse transcriptase (Invitrogen #18080051), hexameric random primers and dNTPs at 47°C for 2 hr and deactivation at 70°C for 15 min. Quantitative real-time PCR (qRT-PCR) reactions were performed using the StepOne Plus Real-Time PCR system (Applied Biosystems #4376600) with a +0.7°C melt-curve increment. Reactions were performed in triplicate using MicroAmp Optical 96-well plates (Applied Biosystems #N8010560) containing 8 µl cDNA sample, 10 µl v/v Fast SYBR green Master Mix (Applied Biosystems #4385612) and 0.1 µM forward and reverse primers in 2 µl. The primer sequences are listed in Appendix 3. Changes in target gene expression were normalized to the non-POU domain-containing octamer-binding protein (NONO) housekeeping gene. Fold changes in gene expression were determined by $2^{(-\Delta\Delta Ct)}$.

## Methylation-specific PCR

Cells were seeded 48 hr prior to genomic DNA (gDNA) extraction. Genomic DNA was extracted using NucleoSpin Tissue Kit (Macherey-Nagel #740952) according to the manufacturer's protocol. Cells were resuspended in lysis buffer containing 1.35 mg/ml proteinase K. Cell lysate was applied to DNA-binding column and gDNA was eluted after binding and wash silica membranes. DNA bisulfite modification was performed using EZ DNA methylation kit (Zymo Research #D5001) according to the manufacturer's protocol; 500 ng gDNA was incubated with CT conversion reagent for 16 hr at 50°C in dark, to allow non-methylated cytosine (C) residue to be converted to uracil (U). gDNA was applied to DNA-binding column and desulfonated for 15 min at RT. Columns were then washed and gDNA was eluted in 10 µl M-Elution buffer. PCR condition and primer sequence were adapted from *Zhao et al., 2012*. PCR amplification reaction mixture was prepared in 100 µl aliquots containing 2 µl of bisulfite converted gDNA, 200 µM dNTPs (Roche #DNTPM-RO), 1 mM Primer, 1.5 mM $MgCl_2$, 50 mM KCl, 10 mM Tris-HCl pH 8.3, and 1.25 units GoTaq DNA Polymerase (Promega #M3001). The primer sequences are listed in Appendix 3. PCR amplification reaction was performed in the T100 thermalcycler under the following conditions: initial denaturation at 95°C for 10 min, followed by 35 cycles (94°C for 30 s, 55°C for 30 s, and 72°C for 30 s) and reaction deactivation at 70°C for 15 min.

## RNA-seq and analysis

RNA-seq and analysis were described previously (*Chan et al., 2020*). Poly-A selective RNA-seq libraries were prepared using the TruSeq RNA sample preparation kit (Illumina) and sequenced on

Illumina NextSeq 500. HISAT2 (version 2.0.4) for the 75 bp single end reads was used for alignment to the genome (hg19/GRCh37). Reads were counted using feature counts (version 1.6.2) in Galaxy. The differential expression of genes was calculated utilizing the DESeq2 package v1.24.0 and plotted in R. Absolute gene expression was defined determining RPKM. FastQ raw data and processed files are available in the public depository NCBI GEO under accession numbers GSE200479. GSEA was performed according to the Hallmark gene set from the molecular signatures database MSigDBv6.1 (Broad Institute).

## GC/MS metabolomics

After saline wash, cells were quenched by pouring liquid nitrogen into six-well plates and then harvested with ice-cold methanol:chloroform:scyllo-inositol (MeOH:CHCl3 9:1 v/v) containing 3 μM scyllo-inositol as internal standard. The extracts were vortexed for 10 s and incubated on ice for 15 min. By centrifugation at 4°C for 3 min at 16,100 $g$, the supernatant was collected and snap-frozen in liquid nitrogen and stored at –80°C.

The samples were evaporated to dryness by vacuum centrifugation. Prior to GC/MS analysis, samples were derivatized with 25 μl 3% (w/v) methoxyamine in pyridine (Sigma, #226904/270970) for 60 min at 37°C with mixing at 750 rpm, followed by trimethylsilylation with 25 μl BSTFA+1% TMCS (Thermo, #38831) for 60 min at 37°C with mixing at 750 rpm. The derivatized sample (1 μl) was analyzed using Shimadzu GC/MS-TQ8040 system, running the Shimadzu Smart Metabolites NRM database, comprising approximately 475 metabolite targets. Statistical analyses were performed using Student's t-test following log transformation and median normalization. Metabolites were considered to be significant if their adjusted p-values after Benjamini-Hochberg correction were less than 0.05. Further data analysis and enrichment analysis were performed through MetaboAnalyst 4.0.

## Isotope tracing analysis by LC/MS

[3–13C] serine was purchased from Cambridge Isotope Laboratories. Unlabeled serine and cystine were purchased from Sigma. The basal isotope labeling medium was prepared following the standard DMEM formula (Thermo, #52100) except without serine and cystine. BJ-TERT cells were plated in six-well plates in normal DMEM containing 10% FBS. On the following day, cells were transduced pBabe control or myrAKT1 retrovirus. On day 2 post-transduction, cells were selected by 1 μg/ml puromycin for 3 days. Cells were then transfected with 20 nM CBS siRNA or non-targeting control siRNA via a reverse transfection approach. On day 2 post-siRNA transfection, the culture medium was replaced with the basal isotope labeling medium plus unlabeled serine and 10% dialyzed FBS, in the presence and absence of cystine. After cystine depletion/repletion for 2 days, the culture medium was replaced with the basal isotope labelling medium containing 400 μM [3–13C] serine or unlabeled serine and 10% dialyzed FBS, in the absence and presence of cystine for 6 hr. After washing cells with saline, liquid nitrogen was immediately poured into plates. Once liquid nitrogen was evaporated, the plates were stored at –80°C.

To collect and analyze metabolites, we optimized thiol derivatization using *N*-ethylmaleimide (NEM) and LC/MS-based detection as described previously (*Ortmayr et al., 2015*). Plates were transferred from –80°C and the ice-cold extraction buffer containing 50 mM NEM (Sigma-Aldrich, #E3876) in 80% v/v methanol and 20% v/v 10 mM ammonium formate (Sigma-Aldrich, #516961) at pH 7 was added. The mixture was scraped and collected into the microcentrifuge tubes. Samples were mixed at 4°C on a vortex mixer for at 1 hr at 1000 rpm before centrifugation at 20,000 $g$ for 10 min at 4°C. Supernatants were stored on ice and sent for LC-MS analysis as described previously (*Ortmayr et al., 2015*). Briefly, metabolites were separated on an Dionex Ultimate 3000RS HPLC system (Thermo Scientific, Waltham, MA) using an InfinityLab Poroshell 120 HILIC-Z (100 × 2.1 mm, 1.9 μm) HPLC column (Agilent Technologies, Santa Clara, CA) maintained at 25°C with buffer A (20 mM $(NH_4)_2CO_3$, pH 9.0; Sigma-Aldrich) and solvent B (100% acetonitrile) at a flow rate of 300 μl/min. Chromatographic gradient started at 90% B, decreased gradually over 10 min to 65% B, then to 20% B at 11.5 min, stayed at 20% B until 13 min, returned to 90% B at 14 min and equilibrated at 90% B until 20 min. Metabolites were detected by mass spectrometry on a Q-Exactive Orbitrap mass spectrometer (Thermo Scientific) using a heated electrospray ionization source. LC-MS data was collected using full-scan acquisition in positive and negative ion mode utilizing polarity switching. Data processing was performed using Tracefinder application for quantitative analysis (Thermo Scientific). NEM-derivatized thiols were

detected in negative ion mode. Metabolite identification was based on accurate mass, retention time, and authentic reference standards. Peak intensities were normalized with cell numbers, and the statistical analyses were carried out using one-way analysis of variance (ANOVA).

## Bioenergetics analysis using the Seahorse XF96 extracellular flux analyzer

All bioenergetics analyses were performed using the Seahorse Bioscience XF96 extracellular flux analyzer (Seahorse Bioscience, Billerica, MA). Cells were seeded in the Seahorse XF96 96-well plate coated with Cell-Tek ($3.5~\mu g/cm^2$, Corning, NY). After incubation for the indicated time period, cells were washed with the assay media (unbuffered DMEM, 11 mM glucose, 2 mM glutamine, 1 mM sodium pyruvate, adjusted pH to 7.4 with 0.1 M NaOH) before incubation in 180 µl of the assay media and equilibrated in a 37°C non-$CO_2$ incubator for 1 hr. The assay protocol consisted of three repeated cycles of 3 min mixing and 3 min of measurement periods, with OCR and extracellular acidification rate determined simultaneously. Basal energetics were established after three of these initial cycles, followed by exposure to the ATP synthase inhibitor, oligomycin (1 µM) for three cycles, then *p*-trifluoromethoxy- phenylhydrazone (1 µM), which uncouples oxygen consumption from ATP production, was added for a further three cycles. Finally, the mitochondrial complex III inhibitor antimycin A (0.5 µM) and the complex I inhibitor rotenone (0.5 µM) were added for three cycles. At the completion of each assay, the cells were stained with 10 µM Hoechst. Images were analyzed using a Cellomics Cellinsight 1 to determine the cell number per well.

## ROS detection

MitoSOX Red mitochondrial superoxide indicator (5 µM, Invitrogen #M36008) or 2′,7′-dichlorofluorescein diacetate (10 µM; Sigma #35845) was added to cell culture and incubated at 37°C for 1 hr. Cells were trypsinized and harvested before analysis by Canto II.

## H$_2$S production measurement

H$_2$S production was measured using 7-azido-4-methylcoumarin (AzMC) fluorescent dye (Sigma, #802409). The protocol was adapted from Szabo Laboratory with minor modifications (*Szabo et al., 2014*). AzMC reaction master mix consisted of 200 mM Tris-HCl pH 8, 20 mM L-cysteine (Sigma, #30089), 1 mM L-homocysteine (Sigma, #69453), 100 µM pyridoxal 5′-phosphate hydrate (Sigma, #P9255), 20 µM AzMC in H$_2$O was prepared and kept on ice. Cells were washed with cool PBS and then harvested in cell lysis buffer (50 mM Tris-HCl pH 8, 150 mM NaCl, 1% v/v IGEPAL CA-630(Sigma-Aldrich-I3021), 1% v/v Triton-X100). Samples were kept on ice for 1 hr and then centrifuged at 20,000 *g* for 10 min at 4°C before protein quantification by DC protein assay; 400 µg proteins were mixed with 100 µl AzMC reaction master mix and incubated at 37°C in dark for 2 hr. Samples were read at 340 nm excitation and 445 nm emission wavelength using Cytation 3 cell imaging multi-mode reader (BioTek).

## GSH assay

GSH assay was performed using Glutathione Assay Kit (Cayman Chemical #703002) according to the manufacturer's protocol. Briefly, cells were seeded 24 hr before harvesting in 10 cm plate at 80% confluency with equal cell number. One extra plate was prepared for cell number counting after cell harvesting. Cells were washed twice with PBS and lifted using rubber scraper in 1 ml ice-cold PBS. After centrifugation at 400 *g* for 5 min at 4°C, the supernatants were discarded and cell pellets were resuspended in 100 µl MES buffer (50 mM MES, 1 mM EDTA, pH 6). Samples were snapped frozen in liquid nitrogen and then thawed in the ice. The procedures were repeated twice for cell lysis. After centrifugation at 12,000 *g* for 15 min at 4°C, the supernatants (100 µl) were collected without disturbing the pellets; 100 µl of 2.5 M metaphosphoric (MPA, Sigma #239275) was added to the supernatants for deproteination. The solutions were then vortexed and kept at RT for 5 min prior to centrifugation at 3000 *g* for 2 min. Ten µl of 4 M triethanolamine (TEAM, Sigma #T58300) was added into the supernatants collected. The deproteinated and neutralized supernatants were used for the determination of the amount of total GSH by mixing with the assay reagents (provided from the kit) in a 96-well plate. After 25 min incubation in dark, the absorbance at 405 nm was measured using a Benchmark Microplate Reader. Data were normalized with cell number and GSH standard curve.

## Immunofluorescence staining

Cells were seeded at least 48 hr before fixation. One μM MitoTracker-Deep Red FM (Invitrogen #M22426) was added into the medium and incubated for 30 min at 37°C before fixation in 4% paraformaldehyde for 10 min at RT. After washing with ice-cold PBS, cells were incubated with the blocking buffer (5% v/v goat serum, 0.1% v/v Triton X-100 in PBS) for 30 min at 37°C; 100 μl of the CBS antibody (Proteintech #14787-1-AP) diluted in the antibody dilution buffer (1% w/v BSA in PBS) at 1:100 was added onto the cells and covered with a coverslip. Cells were incubated at 4°C overnight. After washing with PBS at RT for 5 min of three times, cells were incubated with the secondary antibody EnVision+System HRP labeled polymer goat-anti-rabbit (Dako-K4003) at RT for 30 min. Cells were washed in PBS at RT for 5 min of three times. The Fluorophore Opal520: Excitation 494 nm; Emission 525 nm, AKOYA #SKU FP1487001KT diluted in the antibody dilution buffer at 1:100 was added onto the cells. After incubation at RT for 10 min, cells were washed in PBS at RT for 5 min of five times in dark. Cells were counterstained with DAPI 0.5 μg/ml in PBS for 5 min at RT followed by PBS wash prior to mounting with VECTASHIELD Antifade Mounting Medium. The images were visualized using Nikon C2 Confocal microscope with Nikon Plan Apo VC 60× oil immersion objective (NA 1.4, Nikon, Japan).

## Tissue microarray

The construction of gastric cancer patient TMA has been described previously (*Wang et al., 2013*). TMA slide composes 120 tumor sections and 63 normal sections from 62 individual gastric cancer patients. These patients underwent gastrectomy from 2000 to 2005 in Changhai Hospital, second Military Medical University, Shanghai, China. All patients have not received any anticancer therapy before surgery. The tissue samples were obtained with patient informed consent and the protocol was approved by Institutional Review Board of Second Military Medical University.

TMA slide embedded in paraffin was baked at 60°C for 1 hr before dewaxing. The slide was incubated in 100% v/v xylene for 3 min of three times followed by 100% v/v ethanol for 1 min of three times, 70% v/v ethanol for 1 min and water wash for 1 min to complete dewax/rehydration procedures. Antigen retrieval was conducted in high pH buffer (Agilent-K8004) using pressure cooker for 45 min. The slide was then washed in 0.1% v/v Tween 20 in TBS buffer for 5 min and blocked in the blocking buffer (2% w/v BSA, 0.2% v/v TX-100, 1% v/v goat serum in PBS) for 1 hr at RT. The CBS antibody (Proteintech #14787-1-AP) diluted in the blocking buffer was applied on top of the slide and covered with coverslip. The slide was incubated at 4°C overnight before wash in 0.1% v/v TBST at RT for 5 min of three times. The secondary antibody EnVision+System HRP labeled polymer goat-anti-rabbit (Dako-K4003) was applied followed by incubation at RT for 30 min. The slide was washed in 0.1% v/v TBST at RT for 5 min of three times. The Fluorophore Opal620 (Excitation 588 nm; Emission 616 nm; Cap Color Amber) from Opal 7-Color Manual IHC kit (PerkinElmer-NEL811001KT) was diluted in 1× Plus Amplification Diluent (PerkinElmer-FP1498) and applied to the samples. The slide was incubated for 10 min and washed in 0.1% v/v TBST at RT for 5 min of five times in dark. It was then counterstained with DAPI and washed in PBS prior to mounting with VECTASHIELD Antifade Mounting Medium. The images were taken using VS120 Virtual Slide Microscopy (OLYMPUS-VS120-L100-W) and analyzed using HALO Image Analysis Software (v2.2.1870.17, Indica Labs, Albuquerque, NM). The same threshold settings were applied to individual patient sections. Total number of cells expressing cytoplasm CBS (Red fluorescence) was divided by total cell number (DAPI) from each section and was shown as %Red Positive. Average cytoplasm fluorescence intensity on tumor tissue section were compared to normal tissue section from the same patient and expressed as fold change relative to normal control.

## Senescence-associated β-galactosidase staining

Ten μM EdU was added into the culture medium 24 hr prior to cell fixation with 2% (v/v) paraformaldehyde, 0.2% (v/v) glutaraldehyde for 5 min at RT. Cells were then incubated with X-Gal staining solution (20 mMcitrate buffer/40 mM $Na_2HPO_4$, pH 6.0, 5 mM potassium ferrocyanide (Sigma-P3289), 5 mM potassium ferricyanide, 150 mM NaCl, 2 mM $MgCl_2$, 1 mg/ml X-Gal (Sigma, #B4252)) at 37°C for 24 hr. After permeabilization with 0.5% (v/v) TritonX-100 (Sigma-T8532) in PBS, EdU was fluorescently labeled with Click-iT EdU AlexaFluor-488 imaging kit (Invitrogen, #C10337) according to the manufacturer's instructions. Cells were counterstained with DAPI. Images were obtained using a fluorescence microscope Olympus BX-61 using a 20× objective. A minimum of 200 cells per sample were counted

and the percentage of EdU or SA-βGal-positive cells are quantified. SAβGal was only counted as positive in the absence of EdU incorporation.

## Colony formation assay

Cells were seeded in six-well culture plates. Media was refreshed every 2 days. At the end of experiments, cells were fixed with 100% (v/v) methanol for 30 min at RT and then stained with 0.1% w/v crystal violet for 30 min at RT. After intensively washing with $H_2O$ and drying plates, images were obtained using ChemiDoc Imaging system. Total colony area, expressed as percentage of cell coverage per well, was determined using the ImageJ plugin Colony Area.

## Anchorage-independent soft agar assay

The anchorage-independent soft agar assay was performed as described by *Borowicz et al., 2014*. Cells were seeded at density of $1 \times 10^4$ per well in 0.4% (w/v) noble agar (Difco-214220) in six-well plate with the basal layer of 0.6% w/v noble agar. One ml of culture medium in the presence or absence of doxycycline was added over the upper layer of agar and replaced twice weekly. Colonies were stained by 0.001% crystal blue for 30 min and then extensively washed with PBS. The number of sizable colonies (diameter >50 μm) were manually counted under transmitted light microscopy for quantitative analysis.

## AGS human gastric cancer xenograft

This study was performed in strict accordance with Australian code for the care and use of animals for scientific purposes. All animal experiments were performed with approval from the Animal Experimentation Ethics Committee at the Peter MacCallum Cancer Centre (Ethics number E626). Mice were maintained in the animal facility of Peter MacCallum Cancer Centre with a relative humidity of approximately 50%, a temperature at 21°C and a 14 hr light and 10 hr dark cycle.

Female NSG mice aged between 8 and 10 weeks were purchased from Peter MacCallum Cancer Centre Animal Facility, Australia. $5 \times 10^6$ AGS human gastric cancer cells transduced with RT3-CBS-puro in 100 μl PBS-Matrigel mixture were implanted into the right flank of mice, using pre-cooled 0.3 ml insulin syringes (BD #230-4533). When tumors reached an average volume of 100 mm$^3$ calculated by $V = (W^2 \times L)/2$, mice were randomized into two groups with one group administered with doxycycline both in the drinking water (0.2% w/v doxycycline hyclate [Sigma # D9891] in 2% sucrose [Sigma #S8501]) and food (600 mg/Kg doxycycline, Specialty Feeds #SF08-026) and the other group administered with the drinking water containing 2% sucrose and normal food. Mice were sacrificed once tumors reached 1200 mm$^3$.

## Statistical analyses

Data is presented as mean ± standard error of the mean (SEM) for three or more biological replicates or ± standard deviation (SD) for less than three biological replicates as indicated in the figure legends. The sample size was not determined based on power calculations. Paired or unpaired Student's t-test was used to analyze the difference between the means of two groups. One-sample t-test was used to compare the mean with a hypothetical value. One-way ANOVA with multiple comparison test was used to analyze the difference within the means of more than two groups. Two-way ANOVA with multiple comparison test was performed to analyze the difference within the means according to two independent variables. Statistical significance was calculated using GraphPad Prism (version 9.3.0) and p-values <0.05 was considered significant.

## Acknowledgements

This work was supported by the National Health and Medical Research Council (NHMRC) of Australia (Project grants #1053792 and #1162052) and Cancer Council Victoria Project Grant #1184873. RBP was supported by Senior Research NHMRC Fellowship #1058586. ES is supported by a Victorian Cancer Agency Mid-Career Research Fellowship (MCRF19007). ZH was supported by a Melbourne International Research Scholarship. We acknowledge the Centre for Advanced Histology and Microscopy and the animal facility at the Peter MacCallum Cancer Centre for their support of this work.

# Additional information

## Funding

| Funder | Grant reference number | Author |
|---|---|---|
| National Health and Medical Research Council | Project grant#1053792 | Richard B Pearson |
| National Health and Medical Research Council | Project grant#1162052 | Richard B Pearson |
| Cancer Council Victoria | project grant#1184873 | Richard B Pearson |
| National Health and Medical Research Council | Fellowship#1058586 | Richard B Pearson |
| Victorian Cancer Agency | Research Fellowship (MCRF19007 | Elaine Sanij |

The funders had no role in study design, data collection and interpretation, or the decision to submit the work for publication.

## Author contributions

Haoran Zhu, Data curation, Formal analysis, Investigation, Methodology, Validation, Writing – original draft; Keefe T Chan, Jian Kang, Conceptualization, Formal analysis, Investigation, Methodology, Project administration, Supervision, Validation, Writing – original draft, Writing – review and editing; Xinran Huang, Carmelo Cerra, Formal analysis, Investigation, Validation; Shaun Blake, Investigation, Methodology; Anna S Trigos, Formal analysis; Dovile Anderson, Formal analysis, Investigation, Methodology; Darren J Creek, David P De Souza, Metta Jana, Methodology, Supervision; Xi Wang, Caiyun Fu, Methodology, Resources; Elaine Sanij, Funding acquisition, Supervision; Richard B Pearson, Conceptualization, Funding acquisition, Project administration, Supervision, Writing – original draft, Writing – review and editing

## Author ORCIDs

Keefe T Chan http://orcid.org/0000-0001-6114-3724
Anna S Trigos http://orcid.org/0000-0002-5915-2952
Richard B Pearson http://orcid.org/0000-0001-5919-5090
Jian Kang http://orcid.org/0000-0001-9998-4975

## Ethics

Human subjects: The construction of Gastric cancer patient TMA has been described previously (Wang et al., 2013). TMA slide composes of 120 tumour sections and 63 normal sections from 62 individual gastric cancer patients. These patients underwent gastrectomy from 2000 to 2005 in Changhai hospital, second Military Medical University, Shanghai, China. All patients have not received any anticancer therapy before surgery. The tissue samples were obtained with patient informed consent and the protocol was approved by Institutional Review Board of Second Military Medical University.

This study was performed in strict accordance with Australian code for the care and use of animals for scientific purposes. All animal experiments were performed with approval from the Animal Experimentation Ethics Committee at the Peter MacCallum Cancer Centre (Ethics number E626). Mice were maintained in the animal facility of Peter Maccallum Cancer Centre with a relative humidity of approximately 50%, a temperature at 21°C and a 14-hour light and 10-hour dark cycle.

## Decision letter and Author response

Decision letter https://doi.org/10.7554/eLife.71929.sa1
Author response https://doi.org/10.7554/eLife.71929.sa2

# Additional files

## Supplementary files

• Transparent reporting form

## Data availability

RNA Sequencing data have been deposited in GEO under accession code GSE200479. All data generated or analysed during this study are included in the manuscript and supporting files. Source data files have been provided for Figure 1B, 2A, 2C, 2F, 4C, 4D, 4F, 6F, 7A, 7C, 7E, 7H; Figure 1-figure supplement 1B, Figure 2-figure supplement 1A, 1D and 1E; Figure 3-figure supplement 1B; Figure 4-figure supplement 1A; Figure 5-figure supplement 1A and Figure 6-figure supplement 1D.

The following dataset was generated:

| Author(s) | Year | Dataset title | Dataset URL | Database and Identifier |
|---|---|---|---|---|
| Zhu H, Kang J, Chan KT, Pearson RB | 2022 | RNA-seq analysis of the effect of loss of Tp53, NF1 or CBS in telomerase-reverse transcriptase (TERT)-immortalized human BJ fibroblasts (BJ-TERT) overexpressing myristoylated AKT1 | https://www.ncbi.nlm.nih.gov/geo/query/acc.cgi?acc=GSE200479 | NCBI Gene Expression Omnibus, GSE200479 |

The following previously published datasets were used:

| Author(s) | Year | Dataset title | Dataset URL | Database and Identifier |
|---|---|---|---|---|
| Cancer Genome Atlas Research Network | 2014 | Comprehensive molecular characterization of gastric adenocarcinoma | http://gdac.broadinstitute.org/runs/stddata__2016_01_28/data/STAD/20160128/ | Broad Institute Genome Data Analysis Center, STAD/20160128/ |
| Ghandi M, Huang F, Jané-Valbuena J, Kryukov G, Lo C, McDonald R, Barretina J, Gelfand E, Bielski C, Li H | 2019 | CCLE 2019 | https://depmap.org/portal/download/?release=CCLE+2019 | Demap portal, CCLE 2019 |
| Ghandi M, Huang F, Jané-Valbuena J, Kryukov G, Lo G, McDonald R, Barretina J, Gelfand E, Bielski C, Li H | 2019 | Fusion | https://depmap.org/portal/download/?release=Fusion | Demap portal, Fusion |
| Ghandi M, Huang F, Jané-Valbuena J, Kryukov G, Lo G, McDonald R, Barretina J, Gelfand E, Bielski C, Li H | 2019 | DNA Copy Number | https://depmap.org/portal/download/?release=DNA+Copy+Number | Demap portal, DNA Copy Number |

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

# Appendix 1

## Appendix 1—key resources table

| Reagent type (species) or resource | Designation | Source or reference | Identifiers | Additional information |
|---|---|---|---|---|
| Gene (*Homo sapiens*) | *CBS* | NCBI | NM_000071.3 | |
| Strain, strain background (*Mus musculus*) | NOD scid gamma mouse | Peter MacCallum Cancer Centre Animal Facility, Australia | | |
| Cell line (*Homo sapiens*) | BJ-TERT human foreskin fibroblast | Provided by Robert Weinberg (Massachusetts Institute of Technology, Cambridge, MA) | | BJ3 human skin fibroblasts expressing telomerase reverse transcriptase. Cultured in DMEM+20 mM HEPES, 17% Medium 199, 15% FBS, and 1% GlutaMAX |
| Cell line (*Homo sapiens*) | IMR90 | Originated from ATCC and obtained from the Garvan Institute of Medical Research, Sydney, Australia | ATCC-CL-186 | Cultured in EMEM+10% FBS, 5 mM sodium pyruvate, 1% non-essential amino acids, and 1% GlutaMAX |
| Cell line (*Homo sapiens*) | AGS | ATCC | ATCC-CRL-1739 | Cultured in DMEM+20 mM HEPES, 10% FBS, and 1% GlutaMAX |
| Cell line (*Homo sapiens*) | Hs 746T | ATCC | ATCC-HTB-135 | Cultured in DMEM+20 mM HEPES, 10% FBS, and 1% GlutaMAX |
| Cell line (*Homo sapiens*) | KATO III | ATCC | ATCC-HTB-103 | Cultured in DMEM+20 mM HEPES, 10% FBS, and 1% GlutaMAX |
| Cell line (*Homo sapiens*) | NCI-N87 | ATCC | ATCC-CRL-5822 | Cultured in RPMI+20 mM HEPES, 10% FBS, and 1% GlutaMAX |
| Cell line (*Homo sapiens*) | SNU1 | ATCC | ATCC-CRL-5971 | Cultured in RPMI+20 mM HEPES, 10% FBS, and 1% GlutaMAX |
| Cell line (*Homo sapiens*) | SNU5 | ATCC | ATCC-CRL-5973 | Cultured in IMDM, 20%FBS, and 1% GlutaMAX. |
| Cell line (*Homo-sapiens*) | GES-1 | Provided by Prof. Caiyun Fu (Zhejiang Sci-Tech University, China) | | Cultured in DMEM+20 mM HEPES, 10% FBS, and 1% GlutaMAX |
| Recombinant DNA reagent | pBabe-puro (plasmid) | *Morgenstern and Land, 1990* | Addgene plasmid #1764 | Retroviral vector as the backbone of all pBabe constructs, used as empty vector control |
| Recombinant DNA reagent | pBabe-puro-HA-myrAKT1 (plasmid) | *Astle et al., 2012* | | pBabe-puro construct expressing HA-tagged myrAKT1 |
| Recombinant DNA reagent | pBabe-puro-HRAS$^{G12V}$ | *Astle et al., 2012* | | pBabe-puro construct expressing HA- tagged HRAS$^{G12V}$, gift from Patrick Humbert |
| Recombinant DNA reagent | pCW57.1-HA-myrAKT1 | *Chan et al., 2020* | | Doxycycline-inducible pCW57.1 construct expressing HA-tagged myrAKT1 |
| Recombinant DNA reagent | pREBIR (TRE3G-dsRed-miRE/shRNA-PGK-eBFP2-IRES-rtTA3) | *Kim et al., 2018* | | Retroviral doxycycline-inducible shRNA expression vector as the backbone of pREBIR constructs |

*Appendix 1 Continued on next page*

*Appendix 1 Continued*

| Reagent type (species) or resource | Designation | Source or reference | Identifiers | Additional information |
|---|---|---|---|---|
| Recombinant DNA reagent | pREBIR-shREN | *Chan et al., 2020* | | Doxycycline-inducible pREBIR construct expressing renilla luciferase sequence |
| Recombinant DNA reagent | pREBIR-sh*CBS*#1 | This paper | | Doxycycline-inducible pREBIR construct expressing shRNA targeting *CBS*. The shRNA hairpins were subcloned from Dharmacon pGIPZ-sh*CBS* (Cat# V3LHS_363331) |
| Recombinant DNA reagent | pREBIR-sh*CBS*#2 | This paper | | Doxycycline-inducible pREBIR construct expressing shRNA targeting *CBS*. The shRNA hairpins were subcloned from Dharmacon pGIPZ-sh*CBS* (Cat# V3LHS_363334) |
| Recombinant DNA reagent | pRT3-puro | This paper | | Retroviral doxycycline-inducible vector which is modified from pREBIR by excising the dsRed/mire cassette and replacing eBFP2 with the puromycin resistance gene |
| Recombinant DNA reagent | pRT3-puro-CBS | This paper | | Retroviral doxycycline-inducible vector pREBIR-puro expressing CBS cDNA which encodes human CBS isoform1 |
| Recombinant DNA reagent | pLNCX2 ER:ras | Masashi Narita (*Young et al., 2009*) | Addgene plasmid #67844 | 4-Hydroxytamoxifen-inducible ER:ras fusion protein as the backbone for pLNCX2 ER constructs |
| Recombinant DNA reagent | pLNCX2 ER:CBS WT-FLAG | This paper | | $HRAS^{V12}$ in pLNCX2 ER;ras was replaced with C-terminally FLAG tagged human CBS isoform 1 |
| Recombinant DNA reagent | pLNCX2 ER:CBS Δ468–551-FLAG | This paper | | $HRAS^{V12}$ in pLNCX2 ER;ras was replaced with C-terminally FLAG tagged human CBS isoform 1 with a deletion of regulatory domain CBSD2 (Δ468–551) |
| Recombinant DNA reagent | FgH1t-puro-*PTEN* gRNA | This paper | | FgH1t-puro was modified from FgH1t-GFP (Marco Herold) to replace GFP with a puromycin resistance gene. Lentiviral vector expressing *PTEN* gRNA sequence (TCATCTGG ATTATAGACCAG) |
| Transfected construct (*Homo sapiens*) | pGIPZ-Non-silencing lentiviral shRNA control | Dharmacon (Horizon Discovery, UK) | Cat#RHS4348 | Lentiviral construct to transfect and express shRNA whose sequence contain no homology to known mammalian genes as the negative control for shRNA experiments |

*Appendix 1 Continued on next page*

*Appendix 1 Continued*

| Reagent type (species) or resource | Designation | Source or reference | Identifiers | Additional information |
|---|---|---|---|---|
| Transfected construct (*Homo sapiens*) | pGIPZ-sh*CBS* #1 | Dharmacon (Horizon Discovery, UK) | Cat# V3LHS_363331 | Lentiviral construct to transfect and express the shRNA targeting human CBS |
| Transfected construct (*Homo sapiens*) | pGIPZ-sh*CBS* #2 | Dharmacon (Horizon Discovery, UK) | Cat# V3LHS_363334 | Lentiviral construct to transfect and express the shRNA targeting human CBS |
| Transfected construct (*Homo sapiens*) | On-Targetplus Non-targeting control pool | Dharmacon (Horizon Discovery, UK) | Cat#D-001810-10-05 | Transfected construct (human) as the negative control for RNAi experiment |
| Transfected construct (*Homo sapiens*) | siRNA to human *CBS* (SMARTpool) | Dharmacon (Horizon Discovery, UK) | Cat#L-008617-00-0005 | Transfected construct (human) |
| Biological samples (*Homo sapiens*) | Tissue microarray slide contains 120 tumor sections and 63 normal sections from 62 individual gastric cancer patients | **Wang et al., 2013** | | These patients underwent gastrectomy from 2000 to 2005 in Changhai Hospital, second Military Medical University, Shanghai, China. All patients have not received any anticancer therapy before surgery |
| Antibody | Anti-p53 (mouse monoclonal) | Santa Cruz | Cat#sc-126 | WB (1:1000) |
| Antibody | Anti-AKT (rabbit monoclonal) | Cell Signaling Technology | Cat#4691 | WB (1:1000) |
| Antibody | Anti-HA (mouse monoclonal) | In-house | Cat#12CA5 | WB (1:2000) |
| Antibody | Anti-Ras (mouse monoclonal) | Santa Cruz | Cat#sc-520 | WB (1:500) |
| Antibody | Anti-Cyclin A (mouse monoclonal) | Santa Cruz | Cat#sc-751 | WB (1:1000) |
| Antibody | Anti-Phospho-RB (S807/811) (rabbit monoclonal) | Cell Signaling Technology | Cat#8,561 | WB (1:1000) |
| Antibody | Anti-RB (mouse monoclonal) | BD Pharmingen | Cat#544136 | WB (1:1000) |
| Antibody | Anti-Phospho-AKT(S473) (rabbit monoclonal) | Cell Signaling Technology | Cat#4058 | WB (1:1000) |
| Antibody | Anti-p21(rabbit monoclonal) | Cell Signaling Technology | Cat#2947 | WB (1:1000) |
| Antibody | Anti-p16 (mouse monoclonal) | BD Pharmingen | Cat#550834 | WB (1:1000) |
| Antibody | Anti-CBS (rabbit monoclonal) | Proteintech | Cat#14787–1-AP | WB (1:1000), IF(1:100) |
| Antibody | Anti-PTEN (rabbit monoclonal) | Cell Signaling Technology | Cat#9559 | WB (1:1000) |
| Antibody | Anti-SLC7A11 (rabbit monoclonal) | Cell Signaling Technology | Cat#12691 | WB (1:1000) |
| Antibody | Anti-CTH (rabbit monoclonal) | Cell Signaling Technology | Cat#19689 | WB (1:1000) |
| Antibody | Anti-IL-1α (Goat polyclonal) | R&D Systems | Cat#AF-200-NA | WB (1:1000) |
| Antibody | Anti-IL-1β (mouse monoclonal) | R&D Systems | Cat#MAB201100 | WB (1:1000) |

*Appendix 1 Continued on next page*

*Appendix 1 Continued*

| Reagent type (species) or resource | Designation | Source or reference | Identifiers | Additional information |
|---|---|---|---|---|
| Antibody | Anti-IL-8 (mouse monoclonal) | R&D Systems | Cat#MAB208100 | WB (1:1000) |
| Antibody | Anti-IL-6 (goat polyclonal) | R&D Systems | Cat#AB-206-NA | WB (1:1000) |
| Antibody | Anti-OXPHOS (mouse monoclonal) | abcam | Cat#ab110413 | WB (1:1000) |
| Antibody | Anti-FLAG (mouse monoclonal) | Sigma-Aldrich | Cat#F3165 | WB (1:1000) |
| Antibody | Anti-β-Actin conjugated to HRP (mouse monoclonal) | Sigma-Aldrich | Cat#A3854 | WB (1:10,000) |
| Antibody | Anti-Vinculin conjugated to HRP (mouse monoclonal) | Santa Cruz | Cat# sc-73614HRP | WB (1:2000) |
| Antibody | Goat anti-rabbit IgG (H+L) HRP conjugate (goat polyclonal) | Bio-Rad Laboratories | Cat# 170-6515 | WB (1:2000) |
| Antibody | Goat anti-mouse IgG (H+L) HRP conjugate (goat polyclonal) | Bio-Rad Laboratories | Cat#172-1011 | WB (1:2000) |
| Sequence-based reagent | Human *CBS*-Forward | This paper | PCR primers | 5′-GGGGCTGAGATTGTGAGGAC-3′ |
| Sequence-based reagent | Human *CBS*-Reverse | This paper | PCR primers | 5′-CGGTACTGGTCTAGGATGTGA-3′ |
| Sequence-based reagent | Human *CBS*-Forward | *Zhao et al., 2012* | Methylation-specific PCR primers | 5′-CAGAGGATAAGGAAGCCAAG-3′ |
| Sequence-based reagent | Human *CBS*-Reverse | *Zhao et al., 2012* | Methylation-specific PCR primers | 5′-TCCCAATCTTGTTGATTCTGAC-3′ |
| Sequence-based reagent | Human *CBS* methylated-Forward | *Zhao et al., 2012* | Methylation-specific PCR primers | 5′-CGAGATATTGGTCGGCGTC-3′ |
| Sequence-based reagent | Human *CBS* unmethylated-Forward | *Zhao et al., 2012* | Methylation-specific PCR primers | 5′-TTATGAGATATTGGTTGGTGTT-3′ |
| Sequence-based reagent | Human *CBS* unmethylated-Reverse | *Zhao et al., 2012* | Methylation-specific PCR primers | 5′-TACCCCAACTACAACAAAACA-3′ |
| Commercial assay or kit | Click-iT EdU AlexaFluor-488 imaging kit | Invitrogen | Cat#C10337 | |
| Commercial assay or kit | Qproteome mitochondria isolation kit | Qiagen | Cat#37612 | |
| Commercial assay or kit | ISOLATE-II kit | Bioline | Cat#52073 | |
| Commercial assay or kit | SuperScript III reverse transcriptase | Invitrogen | Cat#18080051 | |
| Commercial assay or kit | Fast SYBR green Master Mix | Applied Biosystems | Cat#4385612 | |
| Commercial assay or kit | NucleoSpin Tissue Kit | Macherey-Nagel | Cat#740952 | |
| Commercial assay or kit | EZ DNA methylation kit | Zymo Research | Cat#D5001 | |
| Commercial assay or kit | Glutathione Assay Kit | Cayman Chemical | Cat#703002 | |

*Appendix 1 Continued on next page*

Appendix 1 Continued

| Reagent type (species) or resource | Designation | Source or reference | Identifiers | Additional information |
|---|---|---|---|---|
| Chemical compound, drug | *O*-(Carboxymethyl) hydroxylamine hemihydrochloride (AOAA) | Sigma-Aldrich | Cat#C13408 | Final conc: 30 µM |
| Chemical compound, drug | Doxycycline hyclate | Sigma-Aldrich | Cat#D5207 | Final conc: 1 µg/ml |
| Chemical compound, drug | (Z)–4-hydroxytamoxifen (4-OHT) | Sigma-Aldrich | Cat#H7904 | Final conc: 20 nM |
| Chemical compound, drug | L-serine (3–13C) | Cambridge Isotope Laboratories | Cat#CLM-1572 | Final conc: 400 µM |
| Chemical compound, drug | NEM | Sigma-Aldrich | Cat#E3876 | Final conc: 50 mM |
| Chemical compound, drug | Ammonium formate | Sigma-Aldrich | Cat#516961 | Final conc: 10 mM |
| Chemical compound, drug | Erastin | Sigma-Aldrich | Cat#E7781 | Final conc: 5 µM |
| Chemical compound, drug | Protease K | Thermo Fisher | Cat#25530029 | Final conc: 0.1 µg/ml |
| Software, algorithm | GraphPad Prism | GraphPad Software | Version 9.3.0 | |
| Software, algorithm | Molecular signatures database | Broad Institute | Version 6.1 | |
| Software, algorithm | MetaboAnalyst | https://www.metaboanalyst.ca | Version 4.0 | |
| Other | DAPI stain | Invitrogen | Cat#D1306 | Final conc: 0.5–1 µg/ml |
| Other | MitoTracker Deep Red FM | Invitrogen | Cat#M22426 | Final conc: 1 µM |
| Other | MitoSOX red | Invitrogen | Cat#M36008 | Final conc: 5 µM |
| Other | 2′,7′-Dichlorofluorescein diacetate (H$_2$DCFDH-DA) | Sigma-Aldrich | Cat#35845 | Final conc: 10 µM |
| Other | 7-Azido-4-Methylcoumarin (AzMC) | Sigma-Aldrich | Cat#802409 | Final conc: 20 µM |
| Other | 5-Ethyl-2′deoxyuridine (EdU) | Invitrogen | Cat#A10044 | Final conc: 10 µM |
| Other | X-gal | Sigma-Aldrich | Cat#B4252 | Final conc: 1 mg/ml |

## Appendix 2

## mer shRNA sequences

| shRNA | mir30 shRNA 97-mer (5'–3') | Notes |
|---|---|---|
| REN | TGCTGTTGACAGTGAGCGCAGGAATTATAATGCTTATCTATAGTGAAGCCACAGATGTATAGATAAGCATTATAATTCCTATGCCTACTGCCTCGGA | |
| CBS#1 | TGCTGTTGACAGTGAGCGCGACTCAGTGCGGAACTACATAGTGAAGCCACAGATGTATGTAGTTCCGCACTGAGTCATGCCTACTGCCTCGGA | shRNA sequence derived from Dharmacon pGIPZ-shCBS #V3LHS_363331 |
| CBS#2 | TGCTGTTGACAGTGAGCGCAGACGGAGCAGACAACCTATAGTGAAGCCACAGATGTATAGGTTGTCTGCTCCGTCTATGCCTACTGCCTCGGA | shRNA sequence derived from Dharmacon pGIPZ-shCBS #V3LHS_363334 |

## Appendix 3

## PCR primer sequences

| Genes (*Homo sapiens*) | Forward (5'–3') | Reverse (5'–3') |
|---|---|---|
| *CXCL1* | AGTCATAGCCACACTCAAGAATGG | GATGCAGGATTGAGGCAAG |
| *IL1A* | ACTGCCCAAGATGAAGACCA | CCGTGAGTTTCCCAGAAGAA |
| *IL1B* | GGAGATTCGTAGCTGGATGC | GAGCTCGCCAGTGAAATGAT |
| *IL6* | AGTGAGGAACAAGCCAGAGC | CATTTGTGGTTGGGTCAGG |
| *IL8* | GTCTGCTAGCCAGGATCCAC | GCTTCCACATGTCCTCACAA |
| *GAPDH* | GGACTCATGACCACAGTCCATGCC | ATGACCTTGCCCACAGCCTTGG |
| *CBS* | GGGGCTGAGATTGTGAGGAC | CGGTACTGGTCTAGGATGTGA |
| *CBS* (MSP) | CAGAGGATAAGGAAGCCAAG | TCCCAATCTTGTTGATTCTGAC |
| *CBS*-methylated | CGAGATATTGGTCGGCGTC | CCCCGACTACGACGAAACG |
| *CBS*-unmethylated | TTATGAGATATTGGTTGGTGTT | TACCCCAACTACAACAAAACA |
| *NONO* | CATCAAGGAGGCTCGTGAGA | TGGTTGTGCAGCTCTTCCAT |

