## [Editor Report]

This paper describes a new mechanism of metabolic escape from senescence. In cells undergoing senescence induced by AKT, the enzyme cystathionine β-synthase (CBS) maintains viability in the senescent state. Suppressing CBS results in senescence escape and continued proliferation, through a mechanism involving changes in mitochondrial metabolism.

---

## [Decision Letter]

**Decision letter after peer review:**

Thank you for submitting your article "Cystathionine-β-synthase is essential for AKT-induced senescence and suppresses the development of gastric cancers with PI3K/AKT activation" for consideration by *eLife*. Your article has been reviewed by 3 peer reviewers, one of whom is a member of our Board of Reviewing Editors, and the evaluation has been overseen by a Reviewing Editor and Richard White as the Senior Editor. The following individual involved in review of your submission has agreed to reveal their identity: Isaac Harris (Reviewer #3).

Essential revisions:

1) A more thorough assessment of the altered pathways, preferably including isotope tracing to conclusively demonstrate changes in CBS activity, is required. All three Reviewers commented on this.

2) More convincing evidence of mitochondrial localization of CBS and of the importance of mitochondrial CBS in senescence. See in particular the comments from Reviewer 2.

*Reviewer #1 (Recommendations for the authors):*

In Figure 2F, the text states that p53 is upregulated by AIS, but this is not clear from the western blot.

*Reviewer #2 (Recommendations for the authors):*

The validation of mitochondrial localized CBS in Figure 4 is not convincing, based on the resolution of the images, the few cells presented, and the single region evaluated. Since there may be a significant amount of cytosolic CBS present in these cells, fluorescence imaging is not appropriate to demonstrate mitochondrial localization. In addition, the image analysis cannot definitively prove that CBS has been imported into mitochondria. These analyses should be replaced with western blot analysis of CBS levels in purified mitochondria, followed by a protease protection assay to demonstrate localization of CBS within the mitochondria.

To demonstrate the requirement for mitochondrial CBS, the authors should rescue their knockdown cell lines with version of CBS in which the mitochondrial targeting sequence (MTS) is deleted, and test the effects on oxygen consumption and ROS, escape from AIS, and tumorigenesis. In corresponding experiments, the authors should test if mitochondrial-CBS is sufficient to regulate these properties: this can be done by adding a well-established MTS to CBS which exclusively direct proteins to the mitochondria. Validation of the localization of these constructs should be performed as described above.

Import of cysteine (via the xCT transporter) potentially compensates for the abundance of transsulfuration and glutathione-related metabolites in the setting of CBS loss. The authors can explore this possibility with the use of xCT-targeted inhibitors (e.g., erastin, sulfasalazine).

*Reviewer #3 (Recommendations for the authors):*

1. Cysteine synthesis via the transsulfuration pathway should be measured using isotope-labeled C13-serine/glycine in AIS and control cells in cystine-replete and cystine-depleted media.

2. Additional time points in AIS and control cells should be examined and metabolic flux analysis using isotope-labeled C13-glutamine be performed, as these experiments may provide clarity to the phenotypes observed and better support the conclusions made.

3. In addition to blocking H2S production, aminoxyacetate (AOAA) impairs the activity of numerous transaminases, which have been shown to support cancer cell survival (Son, Nature, 2013) and thus should be discussed in relation to Figure 1.

4. Statistical analysis should be performed on the co-occurrence of CBS deletions/mutations and PI3K/AKT/mTORC1 signaling pathway alterations in human tumor data presented in Figure 5.

5. Overall, the role of CBS (as presented in the study) is somewhat confusing, since Figure 1 concludes that CBS is essential for the survival of AIS cells, yet Figure 2 and Figure 6 demonstrate that loss of CBS promotes escape from AIS.

---

## [Author Response]

Essential revisions:1) A more thorough assessment of the altered pathways, preferably including isotope tracing to conclusively demonstrate changes in CBS activity, is required. All three Reviewers commented on this.

To more thoroughly assess the activity of the transsulfuration pathway in AKT-induced senescent (AIS) cells and the impact of loss of CBS, we performed a stable isotope tracing assay using ([3-13C] L-serine) followed by LC/MS after thiol derivatization with NEthylmaleimide in proliferating cells, AIS cells and AIS cells with CBS knockdown. We performed the experiments in cysteine-replete and depleted conditions. The data is shown in Figure 3A-3E in the revised manuscript. These results conclusively demonstrate that exogenous cysteine is the major source for GSH synthesis and AKT overexpression increases cysteine import and the subsequent GSH abundance. Loss of CBS does not affect intracellular cysteine and GSH level. Therefore, senescence escape upon CBS depletion is mediated by the mechanisms independent of the transsulfuration pathway.

Please see our detailed responses to the question #1 from Reviewer 1, the question #3 from Reviewer 2 and the question #1 from Reviewer 3.

2) More convincing evidence of mitochondrial localization of CBS and of the importance of mitochondrial CBS in senescence. See in particular the comments from Reviewer 2.

We agree that providing additional evidence of mitochondrial localization of CBS will further highlight the importance of mitochondrial CBS in AKT-induced senescence maintenance. We provide new data in the revised manuscript including (1) Figure 4C showing increased CBS expression in mitochondrial fractions isolated from AIS cells; (2) Figure 4D showing Cterminally FLAG-tagged CBS was resistant to the protease treatment of intact mitochondria and only digested when the mitochondrial membrane was solubilized by Triton-X detergent in a protease protection assay. (3) Figure 4F showing that loss of C-terminal CBSD2 motif abrogated CBS mitochondrial localization; These results thus strongly support a mitochondrial localization of CBS.

To evaluate the functional significance of CBS mitochondrial localization on AIS maintenance, we reconstituted wild type or the C-terminal truncated mutant CBS into CBSdepleted AIS escaped cells. Figure 4G and 4H showed that in contrast to the finding that expression of wild type CBS could re-instate senescence, cells expressing C-terminal truncated CBS protein failed to do so. Accordingly, Figure 5D, 5E and 5F showed that expression of wild type CBS in CBS-depleted AIS escaped cells increased mitochondrial oxidative phosphorylation while expression of C-terminal truncated CBS did not affect its activity. Therefore, CBS mitochondrial localization is required for CBS-dependent changes in mitochondrial oxidative phosphorylation. Our results thus strongly support AKT overexpression promotes CBS translocation to mitochondria and induces oxidative stress by increasing oxidative phosphorylation to sustain senescence state.

Please see our detailed response to the questions #1 and #2 from the Reviewer 2.

Reviewer #1 (Recommendations for the authors):In Figure 2F, the text states that p53 is upregulated by AIS, but this is not clear from the western blot.

We have repeated the Western blot for p53 to more clearly show upregulation of p53 in AIS cells (Figure 2F).

Reviewer #2 (Recommendations for the authors):The validation of mitochondrial localized CBS in Figure 4 is not convincing, based on the resolution of the images, the few cells presented, and the single region evaluated. Since there may be a significant amount of cytosolic CBS present in these cells, fluorescence imaging is not appropriate to demonstrate mitochondrial localization. In addition, the image analysis cannot definitively prove that CBS has been imported into mitochondria. These analyses should be replaced with western blot analysis of CBS levels in purified mitochondria, followed by a protease protection assay to demonstrate localization of CBS within the mitochondria.

As the reviewer suggested, we have now performed mitochondrial fractionation and analysed CBS expression in the mitochondria isolated from proliferating cells and AIS cells. Indeed, we observed increased CBS abundance in the mitochondria of AIS cells (Figure 4C in the revised manuscript).

To further validate CBS localization in the mitochondria, we performed a protease protection assay using the mitochondria isolated from cells expressing wild type CBS fused to a Cterminal FLAG tagged (Figure 4D in the revised manuscript). C-terminally FLAG-tagged CBS was present in intact mitochondria and was resistant to protease treatment. It was only digested when the membrane was solubilized by Triton X-100 detergent, in a manner similar to that of mitochondrial ATP synthase F1 subunit α ATP5A protein. These results thus strongly support a mitochondrial localization of CBS.

To demonstrate the requirement for mitochondrial CBS, the authors should rescue their knockdown cell lines with version of CBS in which the mitochondrial targeting sequence (MTS) is deleted, and test the effects on oxygen consumption and ROS, escape from AIS, and tumorigenesis. In corresponding experiments, the authors should test if mitochondrial-CBS is sufficient to regulate these properties: this can be done by adding a well-established MTS to CBS which exclusively direct proteins to the mitochondria. Validation of the localization of these constructs should be performed as described above.

Human CBS contains a N-terminal heme domain, a catalytic core, and two CBS motifs (CBSD1 and CBSD2) in C-terminal regulatory domain. A non-canonical mitochondrial targeting signal has been reported to reside within the C-terminal CBSD2 motif (Teng et al., PNAS 2013). To confirm the CBSD2 motif is required for CBS residing on mitochondria, we expressed wild type CBS and mutants with deletion of the heme-binding domain (Δ1-70) or C-terminal regulatory domain CBSD2 (Δ468-551) in BJ-TERT cells depleted of endogenous CBS. Consistent with previous finding (Teng et al., PNAS 2013), loss of CBSD2 motif abrogated CBS mitochondrial localization Author response image 1.

**Author response image 1. sa2fig1:** 

To test sufficiency, we also generated CBS mutants with synthetic mitochondrial targeting sequences (Chin RM et al., Cell Reports 2018; Kang YC et al., Experimental and Molecular Medicine 2018) but found they did not express (Lane 10 and 11 in Author response image 2).

To evaluate the functional significance of mitochondrial localization of CBS-dependent on AIS maintenance, we reconstituted CBS-depleted AIS escaped cells with wild type CBS or the CBSD2 truncation mutant. Expression of wild type CBS prevented AIS escape as evidenced by a decrease of EdU positive cells and increase of SA-βGal-positive cells. In contrast, cells expressing the C-terminally truncated CBS protein escaped from AIS (Figure 4G and 4H in the revised manuscript), demonstrating that mitochondrial localization of CBS is required to maintain AIS.To further confirm that in AIS cells, CBS-stimulated oxidative phosphorylation requires mitochondrial localization, we performed Seahorse analysis in AIS cells expressing either wild type or C-terminal truncation mutant CBS protein (Figure 5C in the revised manuscript). Reconstitution with wild type CBS rescued basal OCR and ATP production levels in CBSdepleted AIS cells. In contrast, AIS cells expressing the C-terminal truncated CBS protein failed to restore basal OCR and ATP production (Figure 5E and 5F). Collectively, our results strongly support the concept that AKT overexpression promotes CBS translocation to mitochondria and subsequently increases energy metabolism to sustain the senescence state.

Import of cysteine (via the xCT transporter) potentially compensates for the abundance of transsulfuration and glutathione-related metabolites in the setting of CBS loss. The authors can explore this possibility with the use of xCT-targeted inhibitors (e.g., erastin, sulfasalazine).

Indeed, we performed a [3-13C] L-serine tracing analysis by LC/MS in proliferating cells, AKT-induced senescent (AIS) cells, and AIS cells with CBS knockdown in cysteine replete and depleted conditions. The results showed that the abundance of cysteine and GSH was not affected by CBS depletion (Figure 3D and 3E in the revised manuscript). However, deprivation of cysteine from the medium markedly diminished the intracellular cysteine and GSH abundance in AIS cells. Our results support the concept that AIS cells rely on exogenous cysteine to maintain cysteine and GSH pools. Please see the details in our response to the question #1 from Reviewer 1.

Furthermore, as suggested we treated cells with erastin and showed that the viability of both proliferating cells and AIS cells was decreased after treatment with erastin, and further suppressed by CBS knockdown (Figure 3-supplement 1F). Collectively, our results suggest that cells rely on exogenous cysteine for GSH synthesis and AKT overexpression increases cysteine import and the subsequent GSH abundance which is not affected by loss of CBS.

Reviewer #3 (Recommendations for the authors):1. Cysteine synthesis via the transsulfuration pathway should be measured using isotope-labeled C13-serine/glycine in AIS and control cells in cystine-replete and cystine-depleted media.

As the reviewer suggested, we performed a stable isotope tracing assay using [3-13C] Lserine followed by LC-MS analysis in proliferating cells, AIS cells without or with CBS knockdown in cysteine-replete and cysteine-depleted medium. Please see our response to question#1 from the Reviewer 1.

2. Additional time points in AIS and control cells should be examined and metabolic flux analysis using isotope-labeled C13-glutamine be performed, as these experiments may provide clarity to the phenotypes observed and better support the conclusions made.

We agree that measuring metabolic flux would help to clarify the phenotypes regulated by CBS. As [3-13C] L-serine is known to be incorporated into the cellular GSH pool via transulfuration pathway-derived cysteine (Zhu et al., Cell Metabolism 2019), we therefore established the conditions for [3-13C] L-serine isotope tracing experiment and performed LC-MS analysis rather than using C13-glutamine. We demonstrated that GSH abundance was increased in AIS cells. CBS knockdown in cysteine-replete medium diminished cystathionine synthesis but did not affect the abundance of cysteine and GSH due to an increased uptake of exogenous cysteine (Figure 3A-3E). We believe that these data strongly support the finding that senescence escape upon CBS knockdown is mediated by a transsulfuration pathway-independent mechanism. Therefore, we did not perform C13-glutamine isotope tracing experiment.

3. In addition to blocking H2S production, aminoxyacetate (AOAA) impairs the activity of numerous transaminases, which have been shown to support cancer cell survival (Son, Nature, 2013) and thus should be discussed in relation to Figure 1.

As the reviewer suggested, to include other potential actions of AOAA we have added the following statement in the revised manuscript as:

“this result suggested that H2S as the major metabolite downstream of the transsulfuration pathway has a protective effect on cells expressing hyperactivated AKT although the actions of AOAA on other PLP-dependent enzymes cannot be excluded (Hellmich et al., Antioxid Redox Signal 2015; Szabo et al., PNAS 2013; Asimakopoulou et al., Br J Pharmacol 2013).”

4. Statistical analysis should be performed on the co-occurrence of CBS deletions/mutations and PI3K/AKT/mTORC1 signaling pathway alterations in human tumor data presented in Figure 5.

We have performed a Fisher statistical test to analyse the association of CBS mutations and mutations in PI3K/AKT/mTORC1 signalling pathway in gastric cancer using the TCGA gastric cancer patient data shown in Figure 6—figure supplement 1A (Figure 5—figure supplement 1A in initially submitted manuscript). The data analysis showed that PI3K/AKT/mTORC1 signaling pathway alterations occur in 33% of gastric cancer (142 of 441 samples) with PIK3CA, AKT and PTEN alterations being the most common genetic alterations (Figure 6-supplement 1A). CBS genetic alterations were found in 18 of 441 gastric cancers with 12 cases cooccurring with mutations in PI3K/AKT/mTORC1 pathway. There was a significant association between CBS mutations and PI3K/AKT/mTORC1 pathway activation in gastric cancer (P-value = 0.0031, 95% CI=1.5202 to 14.9139 and odds ratio 4.4905).

5. Overall, the role of CBS (as presented in the study) is somewhat confusing, since Figure 1 concludes that CBS is essential for the survival of AIS cells, yet Figure 2 and Figure 6 demonstrate that loss of CBS promotes escape from AIS.

We have revised the manuscript to clarify the rationale. In Figure 1 we demonstrated that the antioxidant capacity derived from transsulfuration pathway protected AIS cells from cysteine deprivation-induced cell death. While CBS is the key enzyme in the transsulfuration pathway, in Figure 2 we showed that loss of CBS does not affect AIS cell viability but rather allows AIS cells to escape senescence and re-enter the cell cycle. In Figure 3, 4 and 5, we characterized the mechanisms underlying CBS knockdown-mediated AIS escape. In the revised manuscript, we generated new data including Figure 3A-3E by [3-13C] L-serine isotope tracing and LC-MS analysis and Figure 4C-4H and Figure 5D-5F together with our original data to strongly support the finding that the CBS knockdown-mediated AIS escape occurs through suppressing mitochondrial oxidative phosphorylation while retaining antioxidant capacity of the transsulfuration pathway.

We have now summarized our findings in the revised manuscript as:

“Together, these results strongly support the concept that increased oxidative phosphorylation and ROS production sustain AIS status that require CBS. In parallel, AKT activation increases exogenous cysteine import and transsulfuration pathway activity, which consequently stimulates GSH and H2S production, thereby protecting AIS cells from ROS-induced cell death. Importantly, this AKT-dependent increase of the antioxidant capacity is retained in CBS-deficient cells and thus contributes to escape of AIS cells from cell cycle arrest (Figure 4J).”